# TriScore: Post-Hoc Out-of-Distribution Detection with Energy, Boundary Probes, and Transform Consistency

## Abstract

Post-hoc out-of-distribution detection usually relies on a single signal extracted from a frozen classifier, such as confidence, energy, features, or gradients. This dependence becomes fragile under classifier-level adversarial shifts, where the shift can preserve the detector's chosen cue while changing the model's decision behavior. We investigate whether this failure mode can be mitigated by combining complementary detection criteria instead of selecting a single score. We introduce TriScore, a post-hoc OOD detector that combines three ID-calibrated signals from the same frozen backbone: an energy prior, a boundary-fragility score, and a transform-consistency score. These signals are standardized on ID validation data and fused by an input-dependent ID-residual gate, requiring no OOD labels, auxiliary model, or training. Across four ImageNet-scale backbones, TriScore achieves the best mean AUROC among 15 post-hoc baselines under a non-adaptive adversarial-shift protocol, while remaining competitive on standard OOD benchmarks. Ablations show that the gains come from combining complementary cues rather than optimizing a single detector signal. These results support multi-criterion post-hoc detection as a practical route toward more robust OOD detection under classifier-level shifts.

## 1 Introduction

Modern vision systems often reuse a trained backbone across datasets, tasks, and deployment settings. Post-hoc out-of-distribution (OOD) detection methods helps in those cases, where the classifier has already been fixed and the detector must rely on the outputs or internal signals of the existing classifier without retraining it or using OOD data.

Many post-hoc detectors are built around a primary cue, such as softmax confidence (Hendrycks & Gimpel, 2017), logit energy (Liu et al., 2020), activation statistics (Sun et al., 2021; Djurisic et al., 2023), feature distance (Sun et al., 2022), or gradient magnitude (Huang et al., 2021). Each cue emphasizes a different property of the same classifier. A detector built around only one of these cues therefore has little redundancy when a shift happens to preserve that particular statistic, even when other classifier signals are affected.

This lack of redundancy is most visible under classifier-level adversarial shifts. These are not detector-adaptive attacks, but they provide a demanding stress test for post-hoc detectors, since the image remains close to an in-distribution sample while the classifier response changes by design. In this setting, several strong baselines become difficult to distinguish from random guessing on at least one backbone, and the backbone on which a baseline collapses differs across baselines.

TriScore addresses this lack of redundancy. It extracts three signals from the same frozen classifier. The energy branch measures abnormal logit evidence after subtraction of a class-mean logit vector estimated from ID calibration data. The Boundary-Probe branch estimates local decision fragility from a top-$k$ logit margin and the gradient of that margin at a late feature representation. The transform-consistency branch measures whether mild deterministic views lead to stable predictive distributions. These branches are complementary because they fail for different reasons. Energy is cheap and strong on many semantic shifts. Boundary-Probe responds to locally fragile decisions even when the logit magnitude remains plausible. Transform consistency detects predictions that are unstable under benign views.

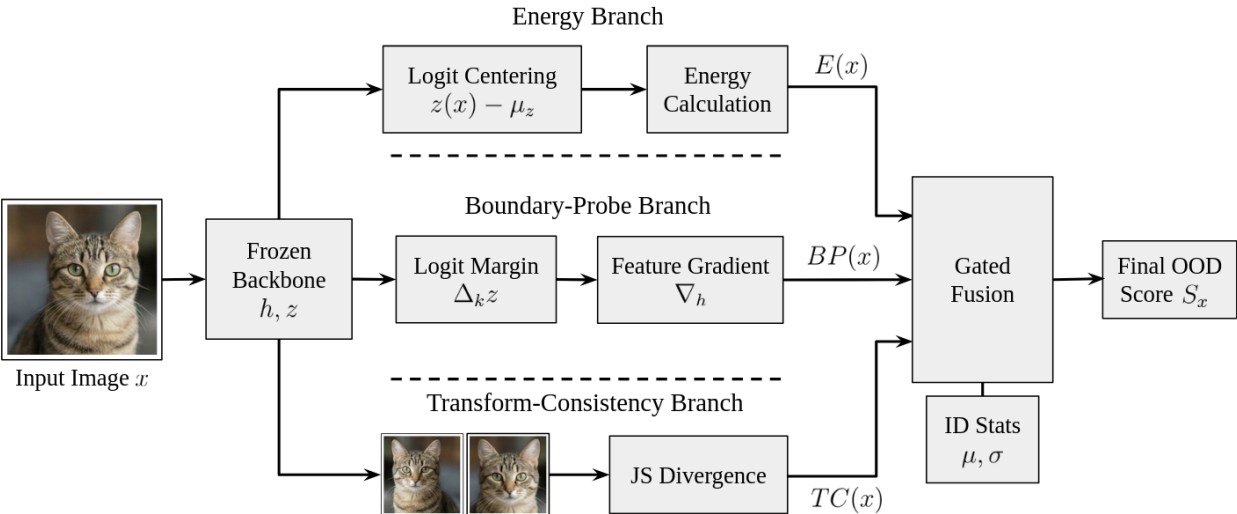

Figure 1: TriScore attaches to a frozen classifier. It computes class-centered energy, Boundary-Probe fragility, and transform consistency, then fuses the standardized ID-residuals into one OOD score. Calibration uses ID validation data only.

After the three signals are computed, TriScore fuses them with a calibrated rule. Each branch is calibrated on ID validation data, which determines the typical behavior for that branch. At test time, the final score gives more weight to branches whose standardized score exceeds the ID-typical range. Calibration does not use OOD examples, and the method does not train auxiliary model. An overview of our method is shown in Figure 1.

We evaluate TriScore in the OpenOOD framework on four ImageNet-scale backbones: ResNet-50, RegNet, ViT-B-16, and Swin-T (He et al., 2016; Radosavovic et al., 2020; Dosovitskiy et al., 2021; Liu et al., 2021). The evaluation uses near-OOD and far-OOD datasets (Bitterwolf et al., 2023; Vaze et al., 2022b; Van Horn et al., 2018; Cimpoi et al., 2014; Wang et al., 2022; Zhang et al., 2023), and adversarial-shifts (Goodfellow et al., 2014; Kurakin et al., 2018; Madry et al., 2017; Dong et al., 2018; Zheng et al., 2019; Croce & Hein, 2020; Carlini & Wagner, 2017; Moosavi-Dezfooli et al., 2016; Andriushchenko et al., 2020; Athalye et al., 2018).

TriScore provides a stronger adversarial-shift robustness floor than the individual post-hoc methods. The per-attack analysis shows that this result is not driven by a single attack, and the transformer results show lower cross-attack variance than the closest competitors.

The contribution is therefore a practical post-hoc detector for frozen classifiers, together with a controlled study of how energy, boundary fragility, and consistency under deterministic transforms interact under clean and adversarial shifts. We make the following claims:

- TriScore combines three frozen-backbone signals under ID-only calibration and remains compatible with standard OpenOOD post-hoc evaluation.

- Under our non-adaptive adversarial-shifts, TriScore gives the best mean AUROC among 15 post-hoc baselines on all four ImageNet-scale backbones, and the best mean FPR@95 on three of four.

- The gain is broad across attack families. TriScore wins 27 of 40 attack×backbone AUROC cells and 19 of 40 FPR@95 cells, and it has lower cross-attack AUROC standard deviation than GEN on ViT-B-16 and Swin-T.

- The ablation study shows that the proposed branches improve over their closest predecessor-style alternatives and that the fusion provides an ID-only way to combine them.

## 2 Related Work

Post-hoc OOD detection scores a frozen classifier without parameter changes. The earliest and still widely used baseline is maximum softmax probability, which treats low maximum confidence as OOD evidence (Hendrycks & Gimpel, 2017). ODIN improves this output-space view with temperature scaling and a small input perturbation (Liang et al., 2018). Later work showed that logits often carry stronger OOD information than normalized softmax probabilities. The maximum logit score uses the largest pre-softmax logit (Hendrycks et al., 2022), energy-based detection uses log-sum-exp free energy (Liu et al., 2020), and GEN studies generalized entropy scores at scale (Liu et al., 2023). These methods are efficient and architecture-agnostic, which makes them strong baselines for any post-hoc method.

A second line of work uses internal network information rather than only final logits. GradNorm uses gradients from a uniform-target objective as an OOD signal (Huang et al., 2021). ReAct and ASH modify late activations by clipping or sparsification before scoring (Sun et al., 2021; Djurisic et al., 2023). DICE changes the effective classifier weights at test time (Sun & Li, 2022), and SCALE revisits activation shaping through activation rescaling (Xu et al., 2024). Feature-space methods use the geometry of ID representations. KNN measures nearest-neighbor distance in deep feature space (Sun et al., 2022). ViM combines the residual outside a principal ID feature subspace with the original logits (Wang et al., 2022). NCI uses neural-collapse-inspired class geometry (Liu & Qin, 2025), while CADRef measures class-aware feature discrepancies relative to classifier-weight directions (Ling et al., 2025). TriScore is motivated by the fact that these cues are useful, but each one is incomplete on its own.

The consistency branch is related to test-time augmentation and distributional disagreement. Test-time augmentation often improves prediction or calibration under benign shifts (Shanmugam et al., 2021), and Jensen-Shannon divergence is a standard symmetric comparison between predictive distributions (Lin, 1991). We use this idea in a post-hoc setting. Our branch compares a base prediction with two mild deterministic views and treats distributional instability as OOD evidence.

Several OOD methods alter the classifier during training. Outlier Exposure trains with auxiliary outlier data (Hendrycks et al., 2019), and energy-based losses can also serve as training objectives (Liu et al., 2020). These approaches are highly effective when retraining and auxiliary OOD data are allowed. They are outside our deployment setting, where the classifier is frozen and the detector must be added after training.

Robustness-oriented OOD detection is closest to our motivation. PRO searches locally around each input for a perturbation that minimizes the OOD score, then uses the resulting score as a robustness-aware detector (Chen et al., 2025). TriScore instead uses fixed forward passes and late-feature backward passes rather than iterative test-time search. The adversarial stress tests in our evaluation draw from the standard attack literature, with FGSM, BIM, PGD and momentum variants, AutoAttack and APGD, C&W, DeepFool, Square, and BPDA-based attacks (Goodfellow et al., 2014; Kurakin et al., 2018; Madry et al., 2017; Dong et al., 2018; Zheng et al., 2019; Croce & Hein, 2020; Carlini & Wagner, 2017; Moosavi-Dezfooli et al., 2016; Andriushchenko et al., 2020; Athalye et al., 2018). Certified robust OOD detection has also been studied (Meinke et al., 2022), but our work makes an empirical post-hoc claim rather than a certified one.

We follow current OpenOOD and full-spectrum evaluation practice (Yang et al., 2022a; Zhang et al., 2023; Yang et al., 2022b). For adversarial generation, the evaluation uses the OpenOOD attack-data interface which uses AutoAttack and Foolbox to generate these attacks (Rauber et al., 2017; Croce & Hein, 2020). ImageNet-scale near-OOD evaluation uses SSB-Hard and NINCO (Vaze et al., 2022b; Bitterwolf et al., 2023), and far-OOD evaluation uses iNaturalist, Textures, and OpenImage-O (Van Horn et al., 2018; Cimpoi et al., 2014; Wang et al., 2022; Zhang et al., 2023). Smaller-scale OSR and generalized OOD studies use CIFAR-10, TinyImageNet, and MNIST (Krizhevsky, 2009; Le et al., 2015; LeCun et al., 2002; Yang et al., 2021; Vaze et al., 2022a). Since post-hoc detector behavior changes with architecture, we evaluate both convolutional and transformer backbones, with ResNet, RegNet, ViT, and Swin (He et al., 2016; Radosavovic et al., 2020; Dosovitskiy et al., 2021; Liu et al., 2021).

## 3 Methodology

We consider post-hoc OOD detection for a frozen image classifier $f : \mathbb{R}^{C \times H \times W} \to \mathbb{R}^K$ with logits $z(x) \in \mathbb{R}^K$. TriScore assigns a scalar score $S(x)$, where larger values indicate stronger evidence that $x$ is OOD. The classifier weights are not updated, and calibration uses only an ID validation split. This setting matches common deployment constraints in which a trained classifier is available, but OOD samples and auxiliary outlier data are not.

TriScore combines three calibrated views of the same frozen classifier. The energy branch measures whether the logit vector looks atypical after removing persistent class-wise offsets. The Boundary-Probe branch estimates local decision fragility by combining a top-$k$ margin with the gradient of that margin at a late feature representation. The transform-consistency branch measures whether the predictive distribution changes under two mild deterministic views. These branches are normalized on ID data and fused with an ID-residual gate that gives more weight to whichever branch is most atypical for the current input.

Two assumptions underlie the method:

- **Assumption 1:** The ID calibration split $\mathcal{D}_{\text{ID}}$ should represent the deployment-time ID distribution, since it determines the branch means, standard deviations, and quantiles. In implementation we use a small standard-deviation floor $\sigma_{\min} = 10^{-6}$ so that a nearly constant ID branch cannot create unstable standardized scores.

- **Assumption 2:** The transform pool should be mostly label preserving for ID inputs. Horizontal flips and mild crops are usually safe for natural images, but not for all classes, especially classes involving orientation, digits, or readable text. We therefore treat transform consistency as an empirical cue rather than an invariance guarantee.

Implementation follows this decomposition. The same identity, flip, and crop views are used for BP and TC, so their forward passes are shared. The backward pass is used only to obtain $\nabla_h \Delta$ at the chosen late feature representation; no classifier parameter is updated. The offline temperature search evaluates four $(T, \tau)$ pairs on the ID calibration split, with BP cached because it does not depend on either temperature. Additional hook locations, numerical safeguards, default constants, and timing conventions are given in Appendix A.

Algorithm 1 gives the full calibration and scoring procedure. The offline phase estimates temperatures, branch statistics, and ID quantiles. The test-time phase evaluates the three branches and applies the same stored statistics to every input.

Temperatures $T$ and $\tau$ are selected by grid search over $\mathcal{T} \times \mathcal{T}_\tau$. We use $\mathcal{T} = \{0.5, 1.0\}$ and $\mathcal{T}_\tau = \{0.5, 1.0\}$, and choose the pair that minimizes the variance of the equal-weight standardized score on up to 5,000 ID calibration samples. Since each candidate is re-standardized before the variance is measured, this criterion favors temperature pairs for which the branches agree on typical ID data. The criterion does not use OOD labels and does not tune directly for benchmark performance.

### 3.1 Energy on class-centered logits

An energy signal is a scalar summary of the full logit vector. Instead of keeping only the largest softmax probability, it uses log-sum-exp to combine the evidence assigned to all classes. For classifier logits $z(x)$, the standard raw energy score is

$$E_{\text{raw}}(x; T) = -T \log \sum_{k=1}^{K} \exp\left(\frac{z_k(x)}{T}\right) \tag{1}$$

This is the standard energy-based OOD formulation used by EBO (Liu et al., 2020). It is efficient as it only requires the logits already produced by the classifier, and it is stronger than maximum softmax probability in many large-scale settings as it preserves logit-scale information (Hendrycks et al., 2022).

The limitation is that raw logits in a frozen classifier can include persistent class-wise offsets inherited from training. Some classes receive systematically larger logits on ID data even when the evidence for a particular

---

**Algorithm 1** TriScore ID-only calibration and test-time scoring

---

**Require:** Frozen classifier $f$, ID validation set $\mathcal{D}_{ID}$, transform pool $\mathbb{T}_{geo}$, temperature grids $\mathcal{T}, \mathcal{T}_\tau$, quantile levels $p_E, p_{BP}, p_C \in (0, 1)$, softness $s > 0$

**Ensure:** Scalar OOD score $S(x) \in \mathbb{R}$ for each test input $x$

1: **Offline calibration**                                           cache BP once and grid-search E and TC temperatures
2: **for** each $(T, \tau) \in \mathcal{T} \times \mathcal{T}_\tau$ **do**
3:     **for** each $x$ in $\mathcal{D}_{ID}$ **do**
4:         compute $E(x; T)$, $BP(x)$, $TC(x; \tau)$                      $\triangleright$ Eqs. equation 4, equation 8–equation 9, and equation 13
5:     **end for**
6:     z-score each branch using statistics from this $(T, \tau)$
7:     form the equal-weight standardized sum on $\mathcal{D}_{ID}$ and record its variance
8: **end for**
9: $(T^*, \tau^*) \leftarrow \arg\min_{(T, \tau)} \mathrm{Var}$
10: recompute scores with $(T^*, \tau^*)$ and estimate $(\mu_., \sigma_.)$
11: set quantile thresholds $\kappa_E, \kappa_{BP}, \kappa_C$
12: store $(T^*, \tau^*, \mu_., \sigma_., \kappa_.)$
13: **Test-time scoring**                                                    $\mathcal{O}(|\mathbb{T}_{geo}| \cdot (F + B))$ per image
14: given test image $x$ and stored statistics
15: compute raw scores $E(x; T^*)$, $BP(x)$, $TC(x; \tau^*)$
16: standardize to get $E_z(x)$, $S_{BP,z}(x)$, $C_z(x)$
17: compute residuals from standardized scores
    $r_E = \max(0, E_z - \kappa_E)$, $r_{BP} = \max(0, S_{BP,z} - \kappa_{BP})$, $r_C = \max(0, C_z - \kappa_C)$
18: compute weights $w_j(x) = e^{r_j / s} / \sum_i e^{r_i / s}$
19: **return** $S(x) = w_E \, E_z + w_{BP} \, S_{BP,z} + w_C \, C_z$

*Here $F$ is the cost of one forward pass and $B$ is the cost of one backward pass through the chosen late feature representation. With $|\mathbb{T}_{geo}| = 3$, test-time scoring performs 3 forward passes and 3 late-feature backward passes per image. Offline calibration caches BP once per ID sample, while E and TC are recomputed over the temperature grid.*

---

input is not stronger. A raw energy score can therefore mix input-level OOD evidence with reusable class-level logit bias. This is related to the logit-scale calibration effects studied in recent OOD work (Djurisic et al., 2024).

Our addition is to keep the standard energy form but remove the ID-estimated class-wise offset before computing it. The calibration mean is

$$\mu_z = \mathbb{E}_{x \in \mathcal{D}_{ID}}[z(x)] \in \mathbb{R}^K \tag{2}$$

and the centered logits are

$$\tilde{z}(x) = z(x) - \mu_z \tag{3}$$

The final TriScore energy branch is the standard energy score evaluated on these centered logits,

$$E(x; T) = -T \log \sum_{k=1}^{K} \exp\left(\frac{\tilde{z}_k(x)}{T}\right) \tag{4}$$

The energy branch therefore answers whether the whole logit vector has unusual evidence after removing the average ID class bias. The centering step is not intended to make energy a stronger standalone detector than standard energy baselines. Its role is to make the energy signal less tied to reusable ID class offsets before it is combined with other signals.

### 3.2 Boundary-Probe on late features

A boundary signal measures how close the current prediction is to a competing decision region. The basic logit-space definition is a margin between the largest logit and plausible alternatives. Let $z_{(1)}(x) \geq z_{(2)}(x) \geq \cdots \geq z_{(K)}(x)$ denote sorted logits. We use the top-$k_m$ margin

$$\Delta(x) = z_{(1)}(x) - \frac{1}{k_m - 1} \sum_{i=2}^{k_m} z_{(i)}(x), \qquad k_m = 5 \tag{5}$$

A top-$k_m$ average is more stable than a pure top-2 margin in large-class problems as it averages several plausible competitors. The operation is piecewise differentiable, thus, we use the standard autograd subgradient, and exact ties are negligible in floating-point inference.

The standard margin method uses the negative margin as OOD evidence, since a smaller margin indicates a more ambiguous prediction. A standard gradient method, such as GradNorm (Huang et al., 2021), instead uses a gradient magnitude as a sensitivity cue. These baselines isolate two different aspects of boundary behavior.

The limitation is however, the margin-only and gradient-only scores are incomplete. The margin says how much separation exists at the current point, but not how quickly that separation changes. The gradient says how sensitive the margin is, but not whether there is enough separation to begin with. Two inputs with the same margin can have different local fragility, and two inputs with the same gradient norm can have different boundary costs.

Our addition is to combine the margin and its late-feature gradient into a first-order fragility proxy. Let $h(x)$ be the selected late feature representation. For CNNs this is the final convolutional block or the pooled representation feeding the head, and for transformer backbones it is the final block representation used by the classifier head. Gradients are taken only with respect to this representation, with the model in evaluation mode so BatchNorm and LayerNorm statistics are fixed. For a transformed view $t(x)$, define

$$g_t(x) = \nabla_{h_t}\Delta(t(x)), \qquad h_t = h(t(x)) \tag{6}$$

Under the local linear approximation $\Delta(h) \approx \Delta(h_t) + g_t^\top(h - h_t)$, the minimum-norm feature perturbation that sets this scalar margin proxy to zero has norm $|\Delta(t(x))|/\|g_t(x)\|_2$. We use the squared form

$$d_t^2(x) = \frac{\Delta(t(x))^2}{\|g_t(x)\|_2^2 + \varepsilon} \tag{7}$$

The quantity in Eq. equation 7 is a local proxy, not an exact classifier boundary distance. The top-$k_m$ averaged margin is not identical to the argmax boundary, and the first-order approximation can fail when the margin landscape is nonlinear. It follows the same robustness intuition as perturbation-based post-hoc OOD scoring (Chen et al., 2025), while avoiding an iterative inner optimization at test time.

The final Boundary-Probe branch converts this squared proxy into a score where larger values mean greater fragility,

$$BP^{(t)}(x) = -\log\big(d_t^2(x) + \varepsilon\big) \tag{8}$$

higher values indicate smaller first-order boundary cost. The same small $\varepsilon$ stabilizes the denominator in Eq. equation 7 and the logarithm. We average over the deterministic transform pool,

$$BP(x) = \frac{1}{|\mathbb{T}_{\text{geo}}|} \sum_{t \in \mathbb{T}_{\text{geo}}} BP^{(t)}(x) \tag{9}$$

This uses one late-feature backward pass per view and reduces single-view gradient noise.

Boundary-Probe therefore answers whether the classifier is locally fragile after accounting for both the current margin and the margin gradient. It complements confidence scores because it can be large even when the current logits still look plausible.

### 3.3 Consistency under deterministic transforms

A transform-consistency signal measures how much a classifier prediction changes under mild views of the same image. The standard definition starts with a transform pool that is expected to preserve the ID label. We use

$$\mathbb{T}_{\text{geo}} = \{\text{id, flip, crop}_{0.9}\} \tag{10}$$

where $\text{crop}_{0.9}$ is a center crop at 90% scale followed by bilinear resize to $(H, W)$. For temperature $\tau > 0$, the tempered softmax is

$$p_\tau(z)_k = \frac{\exp(z_k/\tau)}{\sum_{j=1}^{K} \exp(z_j/\tau)} \tag{11}$$

We compute $p_0(x) = p_\tau(z(x))$ for the base view and $p_t(x) = p_\tau(z(t(x)))$ for each transformed view.

The standard consistency methods compare either predicted labels or scalar uncertainty values across views. Argmax disagreement checks whether the top class changes. TTA entropy variance checks whether the entropy changes across augmentations (Shanmugam et al., 2021). These are useful as they require only additional forward passes and no retraining.

The issue is that both summaries discard information. Argmax disagreement changes only after the top class flips, it misses smaller but systematic distributional shifts. Entropy variance can remain small when probability mass moves among classes with similar entropy. For OOD detection, these partial summaries can miss instability in the predictive distribution.

Our addition is to compare the full predictive distributions with a symmetric Jensen-Shannon (JS) score (Lin, 1991). Probabilities are clamped at a small floor and renormalized before divergences are computed. For two transformed distributions $\tilde{p}_0$ and $\tilde{p}_t$, the JS divergence is

$$\text{JS}(\tilde{p}_0, \tilde{p}_t) = \tfrac{1}{2}\text{KL}\big(\tilde{p}_0 \big\| \tfrac{\tilde{p}_0 + \tilde{p}_t}{2}\big) + \tfrac{1}{2}\text{KL}\big(\tilde{p}_t \big\| \tfrac{\tilde{p}_0 + \tilde{p}_t}{2}\big) \tag{12}$$

The final transform-consistency branch compares the two non-identity views to the base prediction,

$$C(x;\tau) = \frac{1}{2} \sum_{t \in \{\text{flip}, \text{crop}_{0.9}\}} \text{JS}(\tilde{p}_0(x), \tilde{p}_t(x)) \tag{13}$$

This requires no additional forward passes beyond the identity, flip, and crop views already used by the Boundary-Probe branch.

The transform-consistency branch therefore answers whether the classifier can keep a stable predictive distribution under mild views. It captures information that energy and BP do not directly measure.

### 3.4 ID-residual gated fusion

A fusion signal combines several detector scores into one scalar OOD score that is a weighted sum of branch scores after putting them on a comparable scale. Here the branches have different units. Energy is a logit-scale score, BP is a local fragility score, and TC is a distributional disagreement score.

The standard post-hoc fusion method is to z-score each branch on ID data and then average the standardized scores. We first z-score each branch on $\mathcal{D}_{\text{ID}}$,

$$
\begin{aligned}
E_z(x) &= \frac{E(x;T) - \mu_E}{\sigma_E} \\
S_{\text{BP},z}(x) &= \frac{BP(x) - \mu_{\text{BP}}}{\sigma_{\text{BP}}} \\
C_z(x) &= \frac{C(x;\tau) - \mu_C}{\sigma_C}
\end{aligned}
\tag{14}
$$

The standardized scores express all branches in ID-standard-deviation units, which makes them comparable without OOD labels. The corresponding equal-weight fusion is

$$S_{\text{equal}}(x) = \frac{1}{3}\left(E_z(x) + S_{\text{BP},z}(x) + C_z(x)\right) \tag{15}$$

But equal weights apply the same branch balance to every test input. This is restrictive since different inputs can be abnormal in different ways. Some inputs mainly perturb logit evidence, some mainly create boundary fragility, and some mainly create transform instability. A fixed average cannot react to which branch is most ID-atypical for the current sample.

To address this, we replace the fixed equal weights by a per-sample ID-residual gate. The gate uses ID quantiles to determine which branch is unusually large for the current input. Let $\kappa_E$, $\kappa_{\text{BP}}$, and $\kappa_C$ be the empirical $p_E$, $p_{\text{BP}}$, and $p_C$ quantiles of the standardized branches on $\mathcal{D}_{\text{ID}}$. We use $p_E = 0.90$, $p_{\text{BP}} = 0.70$, and $p_C = 0.70$. Energy is a broad logit prior and receives a more conservative upper-tail threshold, while

BP and TC are fragility cues whose signal often begins closer to the ID bulk. These values are fixed ID-only defaults and are not selected using OOD data. For a test input, residuals above the ID thresholds are

$$r_E(x) = \max\big(0,\, E_z(x) - \kappa_E\big)$$
$$r_{\mathrm{BP}}(x) = \max\big(0,\, S_{\mathrm{BP},z}(x) - \kappa_{\mathrm{BP}}\big) \tag{16}$$
$$r_C(x) = \max\big(0,\, C_z(x) - \kappa_C\big)$$

The final TriScore fusion converts these residuals into weights,

$$w_j(x) = \frac{\exp(r_j(x)/s)}{\sum_{i \in \{E,\mathrm{BP},C\}} \exp(r_i(x)/s)}, \qquad j \in \{E, \mathrm{BP}, C\} \tag{17}$$

and returns

$$S(x) = w_E(x)E_z(x) + w_{\mathrm{BP}}(x)S_{\mathrm{BP},z}(x) + w_C(x)C_z(x) \tag{18}$$

We set the softness parameter to $s = 0.6$. As $s \to 0$, the rule approaches a winner-take-all selector over residuals. As $s \to \infty$, the rule approaches the equal-weight average. In the reported setting, the gate is moderate.

The gated fusion answers which branch is most ID-atypical for the current input, while still retaining information from the other branches.

## 4 Evaluation

The evaluation is designed to test the three-branch score against existing post-hoc detectors and measures its stability across architectures and shifts. We use the OpenOOD framework and compare against 15 post-hoc baselines. The baseline set is restricted to detectors that can be attached to the same frozen classifier, without auxiliary OOD data or retraining, because this is the deployment setting TriScore targets. Within that constraint, the methods cover the main post-hoc families: output and logit scores (MSP, MLS, EBO, ODIN, GEN), activation or classifier-weight shaping (ReAct, ASH, DICE, SCALE), gradient or local-robustness scores (GradNorm, PRO), and feature-geometry scores (KNN, ViM, NCI, CADRef). Most baselines are stock OpenOOD postprocessors; CADRef and PRO are integrated locally because they are recent strong post-hoc baselines not present in the current OpenOOD postprocessor registry, and TriScore and its branch ablations are added as OpenOOD-compatible postprocessors. The main ImageNet-scale evaluation uses ResNet-50, RegNet, ViT-B-16, and Swin-T. These backbones span a residual CNN, a modern convolutional architecture, a plain vision transformer, and a hierarchical transformer, while keeping the comparison small enough to run every method and every attack under the same protocol.

### 4.1 Benchmarks and protocols

Following OpenOOD, ImageNet-1K provides the ID data. Near-OOD evaluation uses SSB-Hard and NINCO, and Far-OOD evaluation uses iNaturalist, Textures, and OpenImage-O. We use this dataset suite because it is the standard ImageNet-scale OpenOOD split and because it separates fine-grained near-OOD semantic shifts from broader far-OOD object, scene, and texture shifts. TriScore estimates its class-mean logits, branch statistics, temperatures, quantiles, and gate parameters from an ID-only calibration subset of at most 5,000 images. OOD or adversarial examples are not used for calibration. Additional CIFAR-10 and open-set recognition checks on CIFAR-10, CIFAR-50, TIN-20, and MNIST-6 are reported in Appendices C.1 and C.2.

The adversarial-shift benchmark follows OpenOOD's ImageNet attack-data interface (Yang et al., 2022a; Zhang et al., 2023), with an extended local suite used for evaluation. Foolbox provides the model wrapper and misclassification-criterion interface for the Foolbox-supported attacks, while the AutoAttack-family entries are generated with AutoAttack. Attacks are generated against the frozen ImageNet classifier with ImageNet labels, and the resulting perturbed ImageNet images are evaluated as shifted inputs against clean ImageNet ID images. The attacker does not optimize TriScore, the branch scores, or the gate.

We use AutoAttack (AA), APGD-CE, C&W, DeepFool, FGSM, masked PGD (M-PGD), PGD, Square, BIM, and PGD-BPDA. This attack set is chosen to cover single-step and iterative gradient attacks, optimization-style attacks, decision-boundary attacks, black-box query attacks, masked/local perturbations, BPDA-style attacks, and an AutoAttack ensemble. It extends the default ImageNet attack-script suite used in OpenOOD with PGD-BPDA and AutoAttack-family attacks in the evaluated adversarial-OOD protocol. The $\ell_\infty$ attacks use $\epsilon = 8/255$, while C&W and DeepFool follow their OpenOOD/Foolbox $\ell_2$ implementations. Appendix A gives the attack objectives, label source, implementation details, calibration split, and baseline-compute conventions.

We report FPR@95 and AUROC. FPR@95 is the false positive rate when the true positive rate is fixed at 95% (lower is better). AUROC is the area under the ROC curve and equals the probability that a randomly chosen OOD sample receives a higher score than a randomly chosen ID sample (higher is better). We focus on AUROC and FPR@95 to remain comparable with OpenOOD.

## 4.2 Adversarial OOD detection

Table 1 reports the mean over the 10 non-adaptive classifier attacks. TriScore obtains the best mean AUROC on all four backbones and the best FPR@95 on RegNet, ViT-B-16, and Swin-T. On ResNet-50, it also has the best mean AUROC, but its FPR@95 is not the best; PRO, MSP, and CADRef have lower FPR@95 in this column. On RegNet, TriScore reaches 72.04 AUROC compared with 71.09 for PRO and 70.29 for EBO. On ViT-B-16, it reaches 63.11 AUROC compared with 61.99 for GEN. On Swin-T, it reaches 65.91 AUROC compared with 65.43 for GEN and 65.24 for DICE.

Table 1: Non-adaptive adversarial-shift detection averaged over 10 classifier attacks for each backbone.

| Method | ResNet-50 | | RegNet | | ViT-B-16 | | Swin-T | |
| | FPR@95 | AUROC | FPR@95 | AUROC | FPR@95 | AUROC | FPR@95 | AUROC |
|---|---|---|---|---|---|---|---|---|
| ASH | 90.48 | 55.68 | 93.38 | 52.16 | 95.25 | 51.13 | 94.09 | 52.11 |
| CADRef | 87.62 | 58.70 | 87.47 | 58.92 | 89.86 | 56.81 | 82.30 | 62.64 |
| DICE | 92.28 | 53.04 | 87.80 | 61.08 | 87.52 | 59.35 | 88.16 | 65.24 |
| EBO | 91.67 | 54.60 | 82.76 | 70.29 | 91.98 | 59.57 | 91.61 | 59.40 |
| GEN | 90.92 | 55.50 | 81.36 | 70.16 | 84.68 | 61.99 | 80.65 | 65.43 |
| GradNorm | 93.36 | 51.51 | 94.63 | 47.04 | 93.07 | 53.13 | 93.58 | 48.55 |
| KNN | 89.15 | 57.96 | 87.33 | 58.88 | 91.37 | 54.96 | 86.38 | 59.24 |
| MLS | 91.51 | 54.79 | 82.95 | 69.28 | 91.76 | 59.32 | 91.41 | 59.69 |
| MSP | **85.24** | 59.64 | 84.49 | 61.40 | 88.64 | 58.43 | 88.59 | 59.99 |
| NCI | 90.20 | 55.98 | 93.34 | 53.45 | 92.28 | 54.71 | 83.82 | 61.90 |
| ODIN | 94.22 | 47.83 | 85.98 | 63.92 | 93.14 | 55.94 | 93.57 | 53.95 |
| PRO | 77.83 | 60.39 | 72.79 | 71.09 | 84.23 | 59.44 | 80.98 | 63.21 |
| ReAct | 89.73 | 56.15 | 87.23 | 62.55 | 89.24 | 58.39 | 87.24 | 61.84 |
| SCALE | 90.24 | 56.36 | 89.28 | 62.34 | 91.91 | 59.10 | 92.09 | 56.80 |
| ViM | 90.70 | 56.57 | 87.53 | 59.83 | 93.35 | 55.24 | 85.27 | 63.47 |
| TriScore | 88.64 | **62.11** | **71.26** | **72.04** | **84.10** | **63.11** | **80.01** | **65.91** |

The per-attack tables in Appendix C.4 show that the aggregate result is not caused by a single attack. Across the 40 attack×backbone cells, TriScore gives the best AUROC in 27 cells. The FPR@95 view is more competitive because PRO is explicitly optimized for perturbation robustness. TriScore still has the best FPR@95 in 19 of 40 cells, compared with 16 for PRO, and it has the best mean FPR@95 on three of four backbones in Table 1. The nearest competitors are attack-specific. NCI is strongest on several ResNet-50 optimization attacks, DICE is strong on parts of Swin-T, and GEN is competitive on several transformer cells. The pattern supports the intended role of TriScore as a broad robustness-floor detector rather than a detector specialized to one attack family.

A second useful statistic is cross-attack variance. On Swin-T, TriScore has a per-attack AUROC standard deviation of 4.5, compared with 5.5 for GEN and 6.9 for DICE. On ViT-B-16, TriScore has standard deviation 5.5, compared with 6.6 for GEN and 5.8 for DICE. Some lower-mean baselines such as PRO or MSP have smaller variance, so the stability claim is relative to the high-AUROC competitors rather than a global minimum. This matters because adversarial OOD deployments do not usually know the attack family in advance. A method with high mean performance and moderate attack-to-attack dispersion can be preferable to a method that wins on one attack and collapses on another.

Under this adversarial-shift protocol, TriScore gives the strongest mean AUROC across the evaluated backbones and the best mean FPR@95 on three of four, with lower cross-attack dispersion than the closest high-AUROC competitor on transformer backbones.

## 4.3 Standard OOD detection

Table 2 reports clean-ID OOD results using standard ImageNet ID examples against the near- and far-OOD datasets. The summary uses the OpenOOD group-mean convention: Near-OOD and Far-OOD are averaged separately, and the reported mean is the average of those two groups.

TriScore is competitive on clean OOD, but it is not the clean-OOD winner. The strongest clean baselines vary by architecture: SCALE is strongest on ResNet-50 in this table, GEN is strongest on RegNet, and CADRef is strongest on ViT-B-16 and Swin-T. The E+BP TriScore variant is a lower-cost clean-OOD alternative, while the final gated detector has better clean FPR@95 on all four backbones in this table.

Table 2: Clean ImageNet OOD detection summary for the four standard backbones. Each cell reports the near/far group mean over NINCO, SSB-Hard, iNaturalist, OpenImage-O, and Textures.

| Method | ResNet-50 | | RegNet | | ViT-B-16 | | Swin-T | |
| --- | --- | --- | --- | --- | --- | --- | --- | --- |
| | FPR@95 | AUROC | FPR@95 | AUROC | FPR@95 | AUROC | FPR@95 | AUROC |
| ASH | 41.45 | 86.96 | 64.65 | 76.43 | 95.61 | 52.38 | 94.88 | 45.55 |
| CADRef | 39.22 | 87.96 | 34.48 | 89.96 | **46.55** | **85.19** | **46.77** | **85.36** |
| DICE | 57.15 | 82.01 | 63.44 | 81.17 | 68.59 | 73.84 | 95.66 | 54.46 |
| EBO | 53.49 | 82.68 | 38.10 | 92.09 | 89.27 | 70.70 | 79.45 | 77.27 |
| GEN | 50.46 | 83.31 | **32.02** | **92.83** | 51.50 | 83.82 | 48.10 | 84.97 |
| GradNorm | 63.38 | 81.61 | 88.28 | 56.86 | 93.68 | 40.51 | 95.22 | 41.53 |
| KNN | 52.50 | 80.64 | 34.65 | 90.33 | 51.20 | 82.46 | 52.94 | 80.49 |
| MLS | 53.02 | 83.02 | 38.74 | 91.86 | 85.74 | 75.92 | 74.29 | 80.23 |
| MSP | 58.56 | 80.63 | 46.66 | 87.13 | 66.76 | 79.78 | 60.11 | 81.53 |
| NCI | 42.23 | 87.10 | 62.05 | 82.28 | 57.49 | 81.30 | 52.54 | 83.02 |
| ODIN | 58.22 | 82.11 | 46.32 | 89.40 | 88.37 | 70.19 | 85.56 | 70.72 |
| PRO | 59.21 | 81.00 | 42.32 | 89.80 | 58.51 | 81.94 | 57.27 | 82.63 |
| ReAct | 46.52 | 85.52 | 50.85 | 83.62 | 69.21 | 77.48 | 57.83 | 81.93 |
| SCALE | **38.17** | **88.94** | 41.37 | 91.08 | 90.24 | 66.97 | 85.61 | 70.66 |
| ViM | 48.01 | 82.38 | 35.38 | 90.24 | 51.48 | 84.93 | 49.82 | 84.22 |
| TriScore (E+BP) | 55.88 | 80.24 | 38.10 | 90.15 | 74.08 | 77.43 | 67.42 | 79.58 |
| TriScore | 52.14 | 80.01 | 37.85 | 89.28 | 65.99 | 80.00 | 57.82 | 81.37 |

These clean OOD results should therefore be read differently from the adversarial-shift results. TriScore's main advantage is not clean-OOD dominance; it is the stronger adversarial-shift floor in Table 1. For clean OOD, the E+BP variant is a practical lower-cost TriScore choice when one wants to avoid the extra TC computation and rely only on energy and boundary cues.

The same table also shows a stability advantage that is easy to miss from mean rankings alone. Several baselines have a severe architecture-specific failure on at least one backbone. ASH falls to 52.38 AUROC on ViT-B-16, GradNorm to 56.86 on RegNet, DICE to 54.46 on Swin-T, and SCALE to 66.97 on ViT-B-16. TriScore remains at or above 80 AUROC on all four standard backbones in the clean summary. FPR@95 shows the same stability pattern but not dominance. TriScore stays below 66 FPR@95 on all four backbones, while ASH, GradNorm, DICE, and SCALE each exceed 90 FPR@95 on at least one backbone. This cross-architecture floor is a useful property for post-hoc deployment, where the detector is often attached after the classifier architecture has already been chosen.

TriScore is not the clean-OOD winner, but it remains competitive and avoids the most severe architecture-specific failures. For clean-OOD deployment, the E+BP variant is the lower-cost TriScore choice, while the final gated detector is the main TriScore variant used for adversarial-shift robustness.

Full per-dataset ImageNet baseline results are reported in Appendix C.3, and gated and equal-weight TriScore variants are separated in Appendices B–C.8.

### 4.4 Runtime comparison

Table 3 reports wall-clock timings on a CIFAR-10 benchmark with a single NVIDIA H100 GPU. CIFAR-10 is used here as a manageable runtime benchmark, not as a claim about ImageNet-scale latency. TriScore completes in 65.4 seconds including 3.3 seconds of ID-calibration setup. It is slower than single-forward scores such as MLS and EBO, and also slower than the measured CADRef implementation. It is faster than GEN and KNN, and slightly faster than PRO in this benchmark. PRO has negligible postprocessor setup in the table, but its total includes the APS evaluator-initialization sweep and input-gradient scoring. GEN is nominally a one-forward-pass score, but its generalized-entropy implementation has nontrivial per-batch post-processing overhead in this code path, so the table should be read as an implementation timing rather than a pure pass-count benchmark. Appendix A states which baseline setup costs are included.

Table 3: Runtime ablation on CIFAR-10 by method.

| Method | Total (s) | Setup (s) | Setup note |
| --- | --- | --- | --- |
| ASH | 40.5 | 0.0 | - |
| CADRef | 32.5 | 7.0 | ID-train sweep. |
| DICE | 23.5 | 4.3 | ID-train sweep. |
| EBO | 26.9 | 0.0 | - |
| GEN | 84.8 | 0.0 | - |
| GradNorm | 46.7 | 0.0 | - |
| KNN | 380.4 | 4.5 | ID-train sweep. |
| MLS | 19.2 | 0.0 | - |
| MSP | 25.6 | 0.0 | - |
| NCI | 41.9 | 7.1 | ID-train sweep. |
| ODIN | 39.1 | 0.0 | - |
| PRO | 69.4 | 0.0 | - |
| ReAct | 37.8 | 0.9 | ID-val sweep. |
| SCALE | 40.7 | 0.0 | - |
| ViM | 46.1 | 6.6 | ID-train sweep. |
| TriScore | 65.4 | 3.3 | ID-val sweep. |

TriScore is a mid-cost post-hoc detector in this implementation. It is more expensive than single-forward scores, but much cheaper than KNN and slightly faster than the measured PRO path.

## 5 Limitations

The results should be read in the scope of the evaluated post-hoc setting. TriScore is designed for frozen classifiers, ID-only calibration, and the non-adaptive classifier-level adversarial-shift protocol described in Section 4.1. It is not a certified detector and does not evaluate an attacker that directly optimizes the TriScore score. An adaptive detector-evasion study would require differentiating through the transform pool, the Boundary-Probe gradient computation, and the ID-residual gate. This is a future benchmark, but it is outside the protocol used for the present comparison.

The method is strongest when the deployment setting rewards redundancy across logit energy, local boundary fragility, and transform consistency. Clean OOD benchmarks do not always require all three cues. In the latest branch-combination ablation, the ungated equal-weight E+BP+TC variant has the strongest clean aggregate, while E+BP remains a lower-cost alternative when the extra transform-consistency computation is undesirable. The final gated TriScore should be preferred when robustness to the evaluated adversarial-shift protocol is the primary objective.

The branch behavior is also architecture dependent. On ResNet-50, the energy signal is weak under adversarial shift, and BP-only is better than the full combination. On RegNet, ViT-B-16, and Swin-T, the full combination is more effective. These results suggest that a practical implementation should include a lightweight branch diagnostic before deployment. The current paper uses the ablation evidence to state this rule manually. An automatic ID-only diagnostic for deciding between BP, E+BP, and E+BP+TC remains future work.

All main tables report a single calibration run with fixed ID-only hyperparameters. This follows the OpenOOD comparison style used in the experiments, but close 1–2 AUROC margins should not be inter-

preted as statistically decisive without repeated calibration seeds or bootstrap intervals. The same caution applies to the quantile levels and softness parameter. They are fixed defaults rather than OOD-tuned values. A compact sensitivity sweep over the quantiles, softness, and calibration seed would strengthen the empirical picture. A backbone-adaptive softness parameter could also improve the gate, but we do not tune it without OOD validation data. We also do not report threshold-dependent metrics beyond FPR@95, so fixed-threshold deployment should calibrate the operating point on a held-out ID split.

Boundary-Probe is intentionally high for locally fragile decisions, and hard ID examples can also be locally fragile. The ID quantile threshold limits the effect, but it does not prove that BP is independent of classifier confidence. A useful diagnostic is to plot BP against softmax confidence on ID data and to measure how well BP predicts ID misclassification. This diagnostic would clarify when BP is acting as an OOD cue and when it is mainly a hard-example cue.

TriScore is also more expensive than single-forward detectors. It uses three forward passes and three late-feature backward passes per input. The runtime table reports the resulting implementation cost on CIFAR-10, but ImageNet-scale latency, batch-size sensitivity, and optimized fused implementations remain to be measured. For latency-critical settings, E-only or E+BP variants provide lower-cost alternatives.

## 6 Conclusion

TriScore is a post-hoc OOD detector that attaches to a frozen classifier and combines three ID-calibrated signals, namely class-centered logit energy, Boundary-Probe fragility, and Jensen-Shannon transform consistency. The method uses no OOD labels, auxiliary model, or retraining. Its main design principle is redundancy. A single confidence cue can be preserved by a shift or perturbation, while a combination of logit evidence, boundary fragility, and view consistency is harder to satisfy simultaneously.

Across four ImageNet-scale backbones and 15 post-hoc baselines, TriScore obtains the best mean AUROC in the non-adaptive adversarial-shift protocol on all four backbones and the best mean FPR@95 on three of four. It also gives lower cross-attack AUROC variance than GEN on the two transformer backbones, which indicates a more uniform robustness floor than the closest high-AUROC competitor rather than a gain concentrated on one attack. On standard OOD benchmarks, the method remains competitive and avoids several severe architecture-specific failures, although specialized energy-style baselines are often stronger on clean OOD alone.

The ablations clarify where the gains come from. Boundary-Probe improves over margin-only and GradNorm-style alternatives, and JS-TC improves over simpler transform-consistency baselines. Class-centered energy is best viewed as a calibrated logit prior for fusion rather than a standalone replacement for EBO. The branches are partly redundant under clean shift, where the ungated E+BP+TC variant gives the strongest fixed-subset aggregate and E+BP is a lower-cost alternative. They are more complementary under adversarial shift, where the full gated detector is stronger on backbones with a usable energy signal.

The main open directions are adaptive attacks against the full detector, branch diagnostics that automatically select BP, E+BP, or E+BP+TC for a given backbone, transform pools that handle texture-like OOD and smooth optimization-based perturbations, and repeated-seed calibration studies. These extensions would turn TriScore from a strong post-hoc adversarial-shift detector into a more complete deployment framework for robust OOD detection.

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

## A Reproducibility details

This section collects the protocol details that are easy to miss from the tables in the main tables.

For attacks that require labels, we use the ImageNet ground-truth label of the source validation image. The attack objective is classifier misclassification under the frozen backbone. For Foolbox attacks this is the standard misclassification criterion, for AutoAttack variants it is the standard evaluation routine with the true ImageNet labels. After generation, the detector only sees the perturbed image and does not use labels. Table 4 summarizes the per-attack objective, norm and budget, and implementation details.

Table 4: Adversarial-shift protocol used in the ImageNet attack tables.

| Attack | Objective / label source | Norm and budget | Implementation details |
|---|---|---|---|
| FGSM | classifier loss, ImageNet label | $\ell_\infty, \epsilon = 8/255$ | OpenOOD/Foolbox default FGSM path |
| BIM | classifier loss, ImageNet label | $\ell_\infty, \epsilon = 8/255$ | OpenOOD/Foolbox basic iterative attack |
| PGD | classifier loss, ImageNet label | $\ell_\infty, \epsilon = 8/255$ | OpenOOD/Foolbox PGD defaults |
| PGD-BPDA | classifier loss, ImageNet label | $\ell_\infty, \epsilon = 8/255$ | Foolbox LinfPGD with 100 steps through the BPDA model wrapper |
| AA | classifier loss, ImageNet label | $\ell_\infty, \epsilon = 8/255$ | AutoAttack standard evaluation |
| APGD-CE | classifier cross-entropy, ImageNet label | $\ell_\infty, \epsilon = 8/255$ | AutoAttack APGD-CE component |
| Square | classifier loss, ImageNet label | $\ell_\infty, \epsilon = 8/255$ | AutoAttack Square component with default query schedule |
| C&W | classifier misclassification, ImageNet label | $\ell_2$ | Foolbox L2CarliniWagnerAttack, 1000 steps |
| DeepFool | classifier misclassification, ImageNet label | $\ell_2$ | Foolbox L2DeepFoolAttack defaults |
| M-PGD | classifier loss, ImageNet label | masked perturbation | OpenOOD masked PGD, $\epsilon = 1$, $\alpha = 0.01$, 40 steps, patch size 60 |

TriScore uses only an ID calibration subset of at most 5,000 examples from the OpenOOD ID validation loader to estimate class-mean logits, branch means and standard deviations, temperature choices, branch quantiles, and gate parameters. Detector metrics are computed with the OpenOOD ID test loader and the

corresponding OOD loaders. OOD or adversarial images are not used for calibration. Main clean OOD and ablation summaries follow the OpenOOD convention used in our scripts. Near-OOD datasets are averaged as a group, Far-OOD datasets are averaged as a group, and the reported mean averages the two groups. Adversarial summaries average the 10 attack-specific metrics listed in Table 4.

All branch scores are standardized using ID-only statistics. To avoid numerical instability from near-degenerate branches, the implementation applies a small standard-deviation floor of $10^{-6}$ before z-scoring. Probability computations in JS-TC use clamped probabilities for numerical stability. The fixed default hyperparameters are $k_m = 5$, $p_E = 0.90$, $p_{BP} = p_C = 0.70$, softness $s = 0.6$, temperature grid $T, \tau \in \{0.5, 1.0\}$, and transform pool $\{id, flip, crop_{0.9}\}$. These values are selected without OOD labels. Sensitivity sweeps over these constants require additional runs.

Boundary-Probe backpropagates only to a chosen late feature representation and uses the model in evaluation mode, so BatchNorm and LayerNorm statistics are not updated during scoring. The hook is placed on the last task-level representation before the classification head: the final residual/trunk representation for ResNet-50 and RegNet, the final class-token representation for ViT-B-16, the final pooled stage representation for Swin-T. Earlier network parameters are not updated, gradients are used only to compute the score.

Baselines are run through the OpenOOD post-hoc pipeline with their standard ID-only setup. Methods that require ID-train statistics or feature banks (for example KNN, ViM, DICE, NCI) are given their standard OpenOOD setup step. TriScore uses an ID-validation setup step. The implementation is organized as an OpenOOD compatible postprocessor.

## B    Ablation study

The ablation is organized around three specific questions:

- Does each branch improve over the closest predecessor or simpler variant under the setup?

- Which branch subset should be used under clean OOD and adversarial OOD?

- Does the ID-residual gate improve over equal-weight fusion?

### B.1    Branch isolation

Table 5 isolates the branch definitions. In the energy family, class-centered energy is comparable to standard EBO but is not a uniformly stronger standalone detector. This is why we use it as a calibrated energy component inside TriScore rather than presenting it as a replacement for EBO. In the boundary family, Boundary-Probe improves over both margin-only and GradNorm. In the consistency family, JS-TC improves over argmax disagreement and TTA entropy variance, with the largest benefit in FPR@95.

Table 5: Branch-isolation results averaged.

| Family | Score | Reference | Clean | | Adversarial | |
|---|---|---|---|---|---|---|
| | | | FPR@95 | AUROC | FPR@95 | AUROC |
| Energy | EBO | standard energy | 63.23 | 81.82 | 89.50 | 60.97 |
| Energy | E (Ours) | ID-centered energy | 63.36 | 81.27 | 89.59 | 60.77 |
| Boundary | Margin-only | top-$k$ margin only | 50.94 | 83.96 | 87.03 | 58.86 |
| Boundary | GradNorm | gradient baseline | 84.19 | 55.70 | 93.66 | 50.06 |
| Boundary | **BP (Ours)** | margin + late-feature gradient ratio | 48.66 | 84.50 | 86.16 | 59.47 |
| Consistency | Argmax | TTA label instability | 100.00 | 63.28 | 100.00 | 52.71 |
| Consistency | TTA entropy | TTA probability instability | 72.33 | 72.38 | 88.81 | 58.64 |
| Consistency | **TC (Ours)** | Jensen–Shannon TTA consistency | 54.35 | 78.66 | 84.03 | 61.16 |

As reported in Table 5, the proposed BP and TC branches improve over their closest simpler alternatives, while class-centered energy should be treated as a calibrated component for fusion rather than a standalone

replacement for EBO. The detailed clean results are reported by backbone in Tables 17, 18, 19, and 20; the per-attack adversarial results are reported in Tables 21, 22, 23, and 24.

## B.2 Branch combinations

Table 6 isolates branch combinations. On clean OOD, the full equal-weight fusion E+BP+TC and the gated TriScore detector are tied in aggregate FPR@95 after rounding, both reaching 53.5. The equal-weight fusion is slightly better in AUROC, 82.9 versus 82.7. Thus, the clean panel should not be read as evidence that gating gives a material clean-OOD gain. Instead, gating preserves the best clean FPR@95 of the fixed full fusion while remaining essentially comparable in AUROC.

On adversarial OOD, the pattern changes. TriScore gives the best aggregate FPR@95 and AUROC, improving over E+BP by 2.1 FPR@95 points and 1.8 AUROC points, and over E+BP+TC by 6.3 FPR@95 points and 3.9 AUROC points. The per-backbone results show the main exception: BP-only is stronger than TriScore on ResNet-50, where the energy branch is weak. On RegNet, ViT-B-16, and Swin-T, the gated fusion is stronger. This is why we treat E+BP as a lower-cost clean-OOD option, E+BP+TC as the strongest fixed clean-OOD subset, and TriScore as the preferred adversarial-shift variant.

Table 6: Branch-combination ablation averaged.

| Variant | Clean OOD | | Adversarial OOD | |
|---|---|---|---|---|
| | FPR@95 | AUROC | FPR@95 | AUROC |
| E | 65.0 | 80.8 | 89.5 | 61.0 |
| E+TC | 63.0 | 81.5 | 88.9 | 61.3 |
| E+BP | 58.9 | 81.9 | 83.1 | 64.0 |
| BP | 65.8 | 76.8 | 82.5 | 63.2 |
| BP+TC | 63.7 | 78.3 | 83.6 | 61.1 |
| TC | 55.9 | 79.0 | 90.0 | 55.4 |
| E+BP+TC | **53.5** | **82.9** | 87.3 | 61.9 |
| TriScore | **53.5** | 82.7 | **81.0** | **65.8** |

As reported in Table 6, adding boundary and consistency cues to the energy branch substantially improves adversarial robustness, while the ID-residual gate provides an ID-only way to adapt branch weights per input. In the clean aggregate, the ungated E+BP+TC fusion is the strongest fixed subset and is marginally better in AUROC. Under adversarial shift, the gated TriScore row is substantially stronger.

The detailed clean-OOD branch-combination results are reported by backbone in Tables 25, 26, 27, and 28. The per-attack adversarial results are reported in Tables 29, 30, 31, and 32.

## C   Additional Results

### C.1   CIFAR-10 experiments

Table 7 reports CIFAR-10 results with ResNet-18. The CIFAR results are included as additional small-scale evidence and a low-resolution stress test.

Table 7: Clean OOD detection results on CIFAR-10 with ResNet-18.

| Method | Near-OOD tin FPR@95 | Near-OOD tin AUROC | Far-OOD mnist FPR@95 | Far-OOD mnist AUROC | Far-OOD places365 FPR@95 | Far-OOD places365 AUROC | Far-OOD svhn FPR@95 | Far-OOD svhn AUROC | Far-OOD texture FPR@95 | Far-OOD texture AUROC |
|---|---|---|---|---|---|---|---|---|---|---|
| ASH | 88.09 | 75.58 | 67.89 | 84.07 | 67.82 | 85.08 | 83.79 | 70.21 | 87.06 | 74.06 |
| EBO | 62.41 | 88.31 | 15.49 | 96.32 | 56.38 | 89.64 | 30.13 | 92.38 | 60.22 | 88.64 |
| GEN | 51.82 | 88.87 | 18.01 | 95.00 | 45.81 | 89.74 | 24.76 | 92.04 | 45.87 | 89.36 |
| GradNorm | 95.09 | 54.80 | 78.93 | 70.06 | 89.42 | 67.95 | 91.03 | 49.90 | 98.00 | 46.29 |
| MLS | 62.41 | 88.22 | 15.53 | 96.12 | 56.37 | 89.51 | 30.13 | 92.24 | 60.08 | 88.55 |
| MSP | 47.20 | 88.64 | 19.21 | 93.95 | 41.83 | 89.35 | 24.24 | 91.57 | 40.43 | 89.13 |
| ODIN | 84.03 | 80.95 | 15.39 | 96.60 | 76.46 | 83.86 | 68.81 | 84.52 | 83.01 | 83.86 |
| ReAct | 67.62 | 87.70 | 18.42 | 95.38 | 39.82 | 91.40 | 44.14 | 90.01 | 67.36 | 87.27 |
| CADRef | 33.98 | 90.10 | 25.58 | 92.18 | 33.82 | 90.05 | 29.80 | 90.71 | 26.34 | 91.76 |
| DICE | 66.20 | 78.83 | 45.62 | 82.01 | 77.78 | 74.02 | 30.79 | 91.06 | 68.33 | 79.28 |
| KNN | 29.60 | 91.83 | 21.33 | 93.74 | 30.92 | 91.61 | 23.69 | 92.25 | 23.34 | 93.49 |
| NCI | 40.19 | 89.80 | 29.21 | 91.63 | 34.72 | 90.33 | 37.59 | 89.50 | 27.69 | 92.15 |
| PRO | 31.20 | 90.34 | 32.96 | 88.60 | 34.40 | 89.42 | 22.91 | 91.94 | 30.86 | 89.88 |
| SCALE | 79.91 | 83.84 | 70.19 | 86.44 | 78.97 | 84.22 | 82.06 | 80.39 | 80.87 | 84.01 |
| ViM | 38.69 | 90.05 | 20.20 | 94.86 | 39.77 | 90.03 | 19.61 | 93.99 | 18.64 | 95.58 |
| TriScore | 30.56 | 90.49 | 27.43 | 91.53 | 32.47 | 89.96 | 20.25 | 92.73 | 42.24 | 88.02 |

## C.2 Open-set recognition

This additional open-set recognition (OSR) protocol checks whether the detector remains usable when the unknown samples are held-out semantic classes from a smaller classification problem. Here, the detector is not tested against adversarially shifted ImageNet inputs, but against class-disjoint open-set splits CIFAR6, CIFAR50, TIN20, and MNIST6.

Table 8 shows that TriScore stays competitive in this classic OSR setting, but it is not the strongest reason for adopting the method. Averaged over the four OSR splits, TriScore has a mean FPR@95 of 49.95, close to PRO at 48.18 and KNN at 48.98, while its mean AUROC is 83.47 compared with 84.85 for KNN and 84.69 for GEN. TriScore does not lose basic open-set behavior when moved away from the main settings.

Table 8: Open-set Recognition Results.

| Method | CIFAR6 FPR@95 | CIFAR6 AUROC | CIFAR50 FPR@95 | CIFAR50 AUROC | TIN20 FPR@95 | TIN20 AUROC | MNIST6 FPR@95 | MNIST6 AUROC |
|---|---|---|---|---|---|---|---|---|
| ASH | 75.14 | 84.76 | 54.25 | 82.14 | 65.57 | 73.74 | 8.65 | 97.79 |
| EBO | 75.14 | 84.76 | 54.25 | 82.14 | 65.57 | 73.74 | 8.65 | 97.79 |
| GEN | 73.77 | 85.16 | 54.03 | 82.24 | 65.13 | 73.79 | 9.75 | 97.57 |
| GradNorm | 91.92 | 63.61 | 86.28 | 68.13 | 81.54 | 69.65 | 36.16 | 93.50 |
| MLS | 75.14 | 84.74 | 54.14 | 82.10 | 65.13 | 73.64 | 8.77 | 97.74 |
| MSP | 70.35 | 85.51 | 54.92 | 80.48 | 65.02 | 71.66 | 12.41 | 96.08 |
| ODIN | 80.46 | 81.41 | 55.50 | 81.79 | 65.07 | 73.80 | 8.77 | 97.75 |
| ReAct | 78.36 | 82.53 | 62.09 | 80.49 | 70.58 | 73.04 | 26.87 | 95.70 |
| NCI | 87.14 | 82.58 | 53.08 | 82.83 | 66.91 | 73.07 | 39.22 | 91.92 |
| ViM | 82.09 | 83.04 | 57.38 | 77.74 | 67.30 | 72.55 | 8.65 | 97.79 |
| DICE | 69.24 | 83.47 | 56.01 | 81.11 | 64.07 | 74.55 | 22.04 | 92.43 |
| KNN | 58.65 | 87.41 | 53.65 | 83.12 | 68.97 | 72.65 | 14.65 | 96.21 |
| PRO | 59.45 | 87.41 | 54.87 | 80.88 | 65.91 | 71.43 | 12.48 | 96.03 |
| CADRef | 66.34 | 86.16 | 54.63 | 81.90 | 65.85 | 73.35 | 14.23 | 96.44 |
| SCALE | 75.14 | 84.76 | 54.25 | 82.14 | 65.57 | 73.74 | 8.65 | 97.79 |
| TriScore | 68.08 | 86.05 | 52.74 | 80.59 | 67.35 | 70.28 | 11.65 | 96.97 |

### C.3 Clean OOD detection by backbone and dataset

Table 9: Clean OOD detection results on ImageNet with ResNet-50.

| Method | Near-OOD ninco FPR@95 | Near-OOD ninco AUROC | Near-OOD ssb_hard FPR@95 | Near-OOD ssb_hard AUROC | Far-OOD inaturalist FPR@95 | Far-OOD inaturalist AUROC | Far-OOD openimage_o FPR@95 | Far-OOD openimage_o AUROC | Far-OOD textures FPR@95 | Far-OOD textures AUROC |
|---|---|---|---|---|---|---|---|---|---|---|
| ASH | 53.07 | 83.45 | 73.67 | 72.89 | 14.10 | 97.06 | 29.19 | 93.26 | 15.32 | 96.90 |
| EBO | 60.61 | 79.70 | 76.54 | 72.08 | 31.34 | 90.63 | 38.08 | 89.06 | 45.77 | 88.70 |
| GEN | 54.88 | 81.70 | 75.71 | 72.01 | 26.11 | 92.44 | 34.52 | 89.26 | 46.24 | 87.59 |
| GradNorm | 79.50 | 74.02 | 78.23 | 71.90 | 32.01 | 93.89 | 68.46 | 84.83 | 43.24 | 92.05 |
| MLS | 59.50 | 80.41 | 76.19 | 72.51 | 30.59 | 91.16 | 37.87 | 89.17 | 46.11 | 88.39 |
| MSP | 56.84 | 79.95 | 74.48 | 72.09 | 43.34 | 88.41 | 50.14 | 84.86 | 60.90 | 82.43 |
| ODIN | 68.10 | 77.77 | 76.87 | 71.74 | 36.13 | 91.16 | 46.50 | 88.23 | 49.24 | 89.00 |
| ReAct | 55.91 | 81.73 | 77.57 | 73.02 | 16.73 | 96.34 | 32.57 | 91.87 | 29.63 | 92.79 |
| CADRef | 49.16 | 85.31 | 73.67 | 74.02 | 14.33 | 96.90 | 26.52 | 93.93 | 10.21 | 97.93 |
| DICE | 66.96 | 76.01 | 77.97 | 70.13 | 33.40 | 92.54 | 47.86 | 88.26 | 44.23 | 92.04 |
| KNN | 58.39 | 79.63 | 83.35 | 62.56 | 40.78 | 86.42 | 44.26 | 87.04 | 17.33 | 97.09 |
| NCI | 53.90 | 83.46 | 73.31 | 73.90 | 14.35 | 96.95 | 30.96 | 92.99 | 17.24 | 96.64 |
| PRO | 57.93 | 80.70 | 74.61 | 71.83 | 39.14 | 89.47 | 47.84 | 85.77 | 69.47 | 81.99 |
| SCALE | 51.92 | 85.37 | 67.71 | 77.34 | 9.51 | 98.02 | 28.16 | 93.95 | 11.91 | 97.63 |
| ViM | 62.28 | 78.63 | 80.40 | 65.53 | 30.69 | 89.56 | 32.83 | 90.50 | 10.50 | 97.97 |
| TriScore | 53.70 | 80.17 | 76.16 | 69.91 | 31.93 | 87.47 | 38.64 | 84.22 | 47.48 | 83.28 |

Table 10: Clean OOD detection results on ImageNet with RegNet.

| Method | Near-OOD ninco FPR@95 | Near-OOD ninco AUROC | Near-OOD ssb_hard FPR@95 | Near-OOD ssb_hard AUROC | Far-OOD inaturalist FPR@95 | Far-OOD inaturalist AUROC | Far-OOD openimage_o FPR@95 | Far-OOD openimage_o AUROC | Far-OOD textures FPR@95 | Far-OOD textures AUROC |
|---|---|---|---|---|---|---|---|---|---|---|
| ASH | 75.34 | 71.04 | 81.71 | 64.00 | 45.74 | 86.18 | 67.94 | 81.20 | 38.67 | 88.65 |
| EBO | 42.51 | 91.67 | 62.06 | 85.28 | 7.71 | 98.29 | 25.94 | 95.83 | 38.10 | 93.02 |
| GEN | 34.46 | 92.68 | 58.93 | 85.71 | 5.96 | 98.53 | 15.71 | 96.89 | 30.38 | 93.97 |
| GradNorm | 93.22 | 49.01 | 91.59 | 49.41 | 88.28 | 57.00 | 92.57 | 60.44 | 71.59 | 76.11 |
| MLS | 42.78 | 91.56 | 62.49 | 84.83 | 9.10 | 98.05 | 25.72 | 95.70 | 39.69 | 92.82 |
| MSP | 48.49 | 86.85 | 65.38 | 78.28 | 28.12 | 94.67 | 36.30 | 91.96 | 44.74 | 88.48 |
| ODIN | 52.95 | 87.76 | 60.39 | 83.55 | 22.96 | 95.75 | 44.66 | 91.66 | 40.27 | 92.00 |
| ReAct | 59.82 | 80.91 | 73.00 | 73.17 | 21.24 | 94.14 | 43.45 | 89.20 | 41.20 | 87.25 |
| CADRef | 38.92 | 88.66 | 66.58 | 78.57 | 11.35 | 96.96 | 23.81 | 94.78 | 13.48 | 97.18 |
| DICE | 73.30 | 77.53 | 67.91 | 76.89 | 39.49 | 89.39 | 73.86 | 81.13 | 55.46 | 84.89 |
| KNN | 37.36 | 90.99 | 71.31 | 75.73 | 1.83 | 99.42 | 12.99 | 97.55 | 30.10 | 94.93 |
| NCI | 75.58 | 77.25 | 79.72 | 69.84 | 48.39 | 90.38 | 67.39 | 86.79 | 23.55 | 95.88 |
| PRO | 42.49 | 90.22 | 68.45 | 80.27 | 12.61 | 97.41 | 25.75 | 95.27 | 49.14 | 90.38 |
| SCALE | 49.86 | 89.93 | 66.95 | 82.36 | 11.17 | 97.88 | 33.61 | 94.87 | 28.24 | 95.31 |
| ViM | 38.40 | 90.60 | 73.43 | 77.25 | 2.19 | 99.40 | 13.45 | 97.04 | 28.91 | 93.22 |
| TriScore | 40.28 | 89.42 | 64.36 | 80.87 | 13.20 | 95.15 | 23.62 | 93.50 | 33.31 | 91.58 |

Table 11: Clean OOD detection results on ImageNet with ViT-B-16.

| Method | Near-OOD ninco FPR@95 | Near-OOD ninco AUROC | Near-OOD ssb_hard FPR@95 | Near-OOD ssb_hard AUROC | Far-OOD inaturalist FPR@95 | Far-OOD inaturalist AUROC | Far-OOD openimage_o FPR@95 | Far-OOD openimage_o AUROC | Far-OOD textures FPR@95 | Far-OOD textures AUROC |
|---|---|---|---|---|---|---|---|---|---|---|
| ASH | 95.40 | 52.52 | 93.50 | 53.89 | 97.02 | 50.63 | 94.80 | 55.52 | 98.49 | 48.53 |
| EBO | 94.14 | 66.02 | 92.24 | 58.80 | 83.56 | 79.30 | 88.82 | 76.48 | 83.66 | 81.17 |
| GEN | 59.31 | 82.51 | 82.24 | 70.09 | 22.94 | 93.54 | 35.43 | 90.27 | 38.31 | 90.23 |
| GradNorm | 95.81 | 35.60 | 93.62 | 42.96 | 91.16 | 42.42 | 94.53 | 37.82 | 92.25 | 44.99 |
| MLS | 92.97 | 72.40 | 91.52 | 64.20 | 72.94 | 85.29 | 85.82 | 81.60 | 78.94 | 83.74 |
| MSP | 77.28 | 78.11 | 86.41 | 68.94 | 42.40 | 88.19 | 56.19 | 84.86 | 56.46 | 85.06 |
| ODIN | 92.63 | 65.16 | 88.90 | 63.47 | 81.12 | 79.55 | 91.09 | 71.46 | 85.70 | 77.17 |
| ReAct | 78.51 | 75.43 | 90.46 | 63.10 | 48.25 | 86.11 | 57.67 | 84.29 | 55.88 | 86.66 |
| CADRef | 49.05 | 85.17 | 83.57 | 71.10 | 21.10 | 93.84 | 28.33 | 91.56 | 30.95 | 91.35 |
| DICE | 81.10 | 71.67 | 89.77 | 59.05 | 47.91 | 82.51 | 52.56 | 82.23 | 54.79 | 82.21 |
| KNN | 54.73 | 82.25 | 86.22 | 65.97 | 27.74 | 91.46 | 34.82 | 89.86 | 33.23 | 91.12 |
| NCI | 61.27 | 80.83 | 85.42 | 66.61 | 34.64 | 90.19 | 41.56 | 88.54 | 48.71 | 87.90 |
| PRO | 55.80 | 81.92 | 84.80 | 70.03 | 36.34 | 90.45 | 46.00 | 87.70 | 57.80 | 85.59 |
| SCALE | 94.67 | 61.33 | 92.36 | 56.46 | 86.74 | 73.69 | 89.50 | 72.54 | 84.68 | 78.88 |
| ViM | 57.45 | 84.64 | 90.06 | 69.42 | 17.59 | 95.72 | 29.59 | 92.18 | 40.41 | 90.61 |
| TriScore | 77.99 | 79.07 | 88.55 | 67.66 | 37.82 | 88.74 | 55.24 | 85.25 | 53.05 | 85.93 |

Table 12: Clean OOD detection results on ImageNet with Swin-T.

| Method | Near-OOD ninco FPR@95 | Near-OOD ninco AUROC | Near-OOD ssb_hard FPR@95 | Near-OOD ssb_hard AUROC | Far-OOD inaturalist FPR@95 | Far-OOD inaturalist AUROC | Far-OOD openimage_o FPR@95 | Far-OOD openimage_o AUROC | Far-OOD textures FPR@95 | Far-OOD textures AUROC |
|---|---|---|---|---|---|---|---|---|---|---|
| ASH | 94.38 | 47.07 | 95.70 | 45.87 | 94.24 | 46.62 | 93.74 | 45.64 | 96.16 | 41.66 |
| EBO | 79.16 | 78.25 | 87.45 | 68.21 | 61.52 | 85.13 | 80.81 | 79.86 | 84.45 | 78.96 |
| GEN | 48.04 | 85.18 | 79.19 | 72.75 | 20.51 | 94.23 | 32.69 | 90.56 | 44.57 | 88.15 |
| GradNorm | 94.07 | 44.81 | 93.57 | 50.36 | 95.12 | 38.23 | 96.99 | 33.47 | 97.73 | 34.71 |
| MLS | 75.00 | 80.89 | 86.52 | 70.44 | 49.19 | 88.99 | 74.45 | 83.72 | 79.80 | 81.68 |
| MSP | 61.21 | 81.72 | 80.90 | 71.78 | 37.31 | 89.86 | 49.30 | 85.77 | 60.87 | 83.27 |
| ODIN | 86.05 | 69.35 | 86.45 | 66.94 | 79.35 | 78.10 | 89.82 | 68.93 | 85.46 | 72.86 |
| ReAct | 60.29 | 81.97 | 85.02 | 69.30 | 31.55 | 90.03 | 42.81 | 87.66 | 54.66 | 86.99 |
| CADRef | 51.70 | 84.90 | 81.73 | 71.51 | 19.92 | 94.31 | 27.23 | 92.08 | 33.33 | 91.14 |
| DICE | 97.44 | 48.97 | 96.36 | 47.99 | 98.23 | 46.83 | 96.73 | 57.47 | 88.30 | 77.03 |
| KNN | 58.51 | 79.15 | 85.00 | 64.08 | 30.94 | 88.93 | 35.71 | 88.63 | 35.71 | 90.54 |
| NCI | 56.61 | 82.91 | 84.40 | 68.94 | 26.71 | 92.21 | 35.20 | 89.78 | 41.84 | 88.34 |
| PRO | 52.95 | 83.96 | 79.42 | 72.32 | 37.30 | 90.65 | 45.20 | 87.56 | 62.56 | 83.15 |
| SCALE | 84.20 | 72.47 | 88.94 | 64.64 | 74.76 | 78.60 | 87.91 | 71.18 | 91.30 | 68.52 |
| ViM | 61.16 | 81.69 | 88.57 | 68.79 | 17.98 | 94.61 | 26.55 | 92.32 | 29.78 | 92.67 |
| TriScore | 56.98 | 82.47 | 83.21 | 70.51 | 29.52 | 89.81 | 47.46 | 84.97 | 59.63 | 83.95 |

## C.4 Adversarial OOD detection by backbone and attack

Table 13: Adversarial OOD detection results on ImageNet with ResNet-50.

| Method | AA FPR@95 | AA AUROC | APGD-CE FPR@95 | APGD-CE AUROC | CW FPR@95 | CW AUROC | DF FPR@95 | DF AUROC | FGSM FPR@95 | FGSM AUROC | MPGD FPR@95 | MPGD AUROC | PGD FPR@95 | PGD AUROC | SQUARE FPR@95 | SQUARE AUROC | BIM FPR@95 | BIM AUROC | PGD_BPDA FPR@95 | PGD_BPDA AUROC |
|---|---|---|---|---|---|---|---|---|---|---|---|---|---|---|---|---|---|---|---|---|
| ASH | 95.50 | 52.08 | 95.46 | 52.12 | 88.62 | 59.91 | 86.52 | 62.24 | 74.61 | 73.27 | 93.48 | 48.94 | 92.52 | 52.05 | 86.33 | 63.02 | 96.46 | 46.40 | 95.33 | 46.81 |
| EBO | 96.22 | 49.83 | 96.34 | 49.85 | 89.77 | 59.55 | 88.23 | 60.95 | 78.17 | 73.49 | 92.23 | 52.61 | 93.94 | 49.37 | 88.82 | 62.33 | 96.87 | 44.33 | 96.08 | 43.71 |
| GEN | 95.04 | 52.15 | 95.14 | 52.18 | 89.63 | 58.80 | 88.19 | 60.28 | 77.72 | 71.92 | 91.23 | 53.69 | 92.72 | 51.28 | 88.67 | 61.91 | 95.97 | 46.58 | 94.91 | 46.25 |
| GradNorm | 98.18 | 43.96 | 98.19 | 43.94 | 88.78 | 62.20 | 88.21 | 63.44 | 81.20 | 72.42 | 97.72 | 39.93 | 96.03 | 47.21 | 89.48 | 61.85 | 98.11 | 39.40 | 97.75 | 40.75 |
| MLS | 96.15 | 50.31 | 96.28 | 50.32 | 89.60 | 59.67 | 88.12 | 61.04 | 77.65 | 73.41 | 92.10 | 52.74 | 93.82 | 49.56 | 88.55 | 62.58 | 96.82 | 44.43 | 96.02 | 43.87 |
| MSP | 87.31 | 59.69 | 87.33 | 59.67 | 87.50 | 60.27 | 87.04 | 60.97 | 73.08 | 70.85 | 85.72 | 57.16 | 82.67 | 57.37 | 85.08 | 64.47 | 89.52 | 52.32 | 86.57 | 53.60 |
| ODIN | 99.13 | 40.02 | 99.12 | 40.01 | 91.76 | 55.85 | 90.78 | 57.10 | 82.38 | 69.41 | 94.25 | 47.71 | 96.94 | 40.43 | 90.68 | 58.15 | 98.69 | 35.89 | 98.50 | 33.75 |
| ReAct | 92.78 | 53.55 | 92.74 | 53.59 | 90.10 | 57.62 | 87.92 | 59.90 | 77.80 | 71.91 | 89.02 | 55.00 | 91.25 | 52.01 | 87.70 | 61.67 | 94.67 | 48.33 | 93.34 | 47.87 |
| CADRef | 89.08 | 57.20 | 89.17 | 57.25 | 89.42 | 59.99 | 87.47 | 62.08 | 74.90 | 73.63 | 89.14 | 52.64 | 89.14 | 55.88 | 86.59 | 63.49 | 90.64 | 52.45 | 90.62 | 52.42 |
| DICE | 96.85 | 46.53 | 96.97 | 46.54 | 90.15 | 58.50 | 88.88 | 61.99 | 78.74 | 74.50 | 94.07 | 47.67 | 94.49 | 47.58 | 89.08 | 62.11 | 97.05 | 41.59 | 96.53 | 41.33 |
| KNN | 87.73 | 60.29 | 87.67 | 60.34 | 92.90 | 53.68 | 92.28 | 55.26 | 85.65 | 63.19 | 89.12 | 56.31 | 89.26 | 57.19 | 90.96 | 57.49 | 87.26 | 58.49 | 88.66 | 57.33 |
| NCI | 95.36 | 51.79 | 95.38 | 51.82 | 88.58 | 61.17 | 86.58 | 63.33 | 74.01 | 74.52 | 93.11 | 48.57 | 91.98 | 52.18 | 86.05 | 64.03 | 96.43 | 44.92 | 94.91 | 46.51 |
| PRO | 74.17 | 60.29 | 74.17 | 60.18 | 84.75 | 60.30 | 84.12 | 60.90 | 73.02 | 71.61 | 77.95 | 57.06 | 73.52 | 58.24 | 85.74 | 63.02 | 76.60 | 56.39 | 74.29 | 55.93 |
| SCALE | 94.91 | 53.22 | 94.92 | 53.26 | 87.69 | 61.41 | 85.96 | 63.55 | 75.77 | 73.37 | 94.78 | 46.81 | 92.18 | 53.14 | 85.36 | 64.36 | 95.94 | 46.61 | 94.90 | 47.85 |
| ViM | 93.57 | 56.21 | 93.68 | 56.25 | 90.58 | 56.50 | 88.82 | 58.56 | 78.59 | 68.05 | 90.59 | 54.59 | 92.90 | 53.66 | 88.48 | 60.12 | 95.17 | 50.78 | 94.60 | 50.94 |
| TriScore | 92.12 | 63.58 | 92.30 | 63.65 | 89.08 | 58.85 | 87.44 | 60.16 | 74.06 | 71.15 | 88.34 | 61.40 | 89.39 | 61.61 | 87.40 | 62.65 | 94.03 | 58.76 | 92.27 | 59.25 |

Table 14: Adversarial OOD detection results on ImageNet with RegNet.

| Method | AA FPR@95 | AA AUROC | APGD-CE FPR@95 | APGD-CE AUROC | CW FPR@95 | CW AUROC | DF FPR@95 | DF AUROC | FGSM FPR@95 | FGSM AUROC | MPGD FPR@95 | MPGD AUROC | PGD FPR@95 | PGD AUROC | SQUARE FPR@95 | SQUARE AUROC | BIM FPR@95 | BIM AUROC | PGD_BPDA FPR@95 | PGD_BPDA AUROC |
|---|---|---|---|---|---|---|---|---|---|---|---|---|---|---|---|---|---|---|---|---|
| ASH | 95.11 | 49.71 | 95.12 | 49.68 | 90.58 | 57.64 | 90.33 | 58.88 | 92.61 | 52.04 | 96.32 | 43.86 | 92.55 | 54.19 | 92.86 | 54.67 | 94.81 | 48.94 | 93.49 | 52.02 |
| EBO | 80.84 | 71.82 | 80.53 | 71.85 | 86.99 | 66.25 | 87.16 | 66.17 | 79.07 | 73.26 | 83.12 | 71.50 | 84.02 | 68.27 | 83.47 | 70.84 | 80.70 | 72.28 | 81.68 | 70.69 |
| GEN | 78.85 | 71.96 | 78.72 | 71.96 | 86.68 | 65.39 | 86.87 | 65.40 | 77.92 | 72.74 | 80.49 | 72.38 | 83.52 | 67.61 | 80.83 | 71.60 | 78.85 | 72.30 | 80.86 | 70.27 |
| GradNorm | 96.38 | 43.47 | 96.36 | 43.42 | 91.11 | 55.95 | 91.14 | 56.31 | 94.50 | 47.00 | 97.08 | 35.45 | 93.81 | 50.82 | 94.29 | 47.88 | 96.54 | 42.78 | 95.13 | 47.35 |
| MLS | 80.98 | 70.76 | 80.69 | 70.78 | 87.13 | 65.30 | 87.28 | 65.23 | 79.33 | 72.34 | 83.40 | 70.36 | 84.20 | 67.36 | 83.62 | 69.76 | 80.98 | 71.27 | 81.93 | 69.66 |
| MSP | 82.66 | 62.27 | 82.81 | 62.26 | 87.80 | 58.25 | 87.98 | 58.13 | 82.22 | 64.49 | 84.21 | 61.99 | 86.08 | 60.20 | 85.31 | 62.15 | 82.93 | 63.24 | 84.24 | 61.66 |
| ODIN | 86.57 | 63.02 | 86.57 | 63.02 | 87.15 | 63.27 | 87.08 | 63.31 | 81.97 | 67.71 | 86.51 | 63.91 | 86.42 | 62.86 | 85.44 | 65.49 | 86.11 | 63.19 | 85.97 | 63.45 |
| ReAct | 85.95 | 63.83 | 85.84 | 63.83 | 90.86 | 58.50 | 90.32 | 59.99 | 84.91 | 63.70 | 86.65 | 63.80 | 88.16 | 61.53 | 87.32 | 63.09 | 85.52 | 64.06 | 86.76 | 63.20 |
| CADRef | 88.42 | 57.00 | 88.55 | 56.96 | 86.68 | 62.70 | 85.45 | 64.31 | 87.49 | 58.97 | 88.48 | 53.73 | 88.90 | 58.84 | 83.86 | 62.48 | 88.35 | 56.51 | 88.54 | 57.71 |
| DICE | 88.87 | 59.59 | 88.88 | 59.56 | 86.22 | 65.38 | 86.55 | 64.90 | 85.61 | 62.38 | 90.41 | 55.23 | 87.31 | 61.81 | 87.24 | 62.13 | 89.03 | 58.90 | 87.92 | 60.96 |
| KNN | 86.35 | 60.30 | 86.27 | 60.31 | 90.76 | 54.40 | 90.26 | 55.76 | 87.85 | 59.83 | 80.00 | 63.28 | 90.89 | 56.14 | 84.67 | 60.65 | 86.77 | 60.33 | 89.52 | 57.75 |
| NCI | 95.53 | 50.50 | 95.52 | 50.44 | 89.71 | 50.14 | 89.51 | 61.45 | 92.62 | 53.84 | 96.34 | 43.56 | 94.33 | 50.36 | 86.05 | 64.03 | 96.43 | 49.67 | 93.64 | 53.11 |
| PRO | 67.48 | 74.82 | 67.55 | 74.81 | 78.86 | 65.52 | 79.00 | 65.49 | 74.07 | 71.44 | 72.51 | 70.58 | 74.26 | 69.93 | 75.20 | 70.88 | 68.35 | 74.68 | 70.64 | 72.78 |
| SCALE | 90.33 | 61.07 | 90.33 | 61.05 | 88.49 | 64.13 | 88.49 | 64.39 | 86.92 | 64.76 | 91.81 | 57.53 | 89.86 | 62.36 | 88.66 | 64.49 | 89.81 | 61.38 | 89.05 | 62.25 |
| ViM | 86.65 | 61.46 | 86.54 | 61.47 | 91.56 | 55.07 | 90.50 | 56.93 | 87.67 | 59.10 | 78.91 | 66.57 | 90.51 | 56.55 | 86.67 | 61.61 | 86.95 | 61.22 | 89.36 | 58.30 |
| TriScore | 65.18 | 75.36 | 64.97 | 75.36 | 78.99 | 66.67 | 79.20 | 66.85 | 72.27 | 72.07 | 70.80 | 72.39 | 73.95 | 70.57 | 72.66 | 71.54 | 65.75 | 75.81 | 68.85 | 73.76 |

Table 15: Adversarial OOD detection results on ImageNet with ViT-B-16.

| Method | AA FPR@95 | AA AUROC | APGD-CE FPR@95 | APGD-CE AUROC | CW FPR@95 | CW AUROC | DF FPR@95 | DF AUROC | FGSM FPR@95 | FGSM AUROC | MPGD FPR@95 | MPGD AUROC | PGD FPR@95 | PGD AUROC | SQUARE FPR@95 | SQUARE AUROC | BIM FPR@95 | BIM AUROC | PGD_BPDA FPR@95 | PGD_BPDA AUROC |
|---|---|---|---|---|---|---|---|---|---|---|---|---|---|---|---|---|---|---|---|---|
| ASH | 94.49 | 52.91 | 94.40 | 53.02 | 95.79 | 48.37 | 95.74 | 48.55 | 94.37 | 53.67 | 94.47 | 55.91 | 95.70 | 49.01 | 97.41 | 47.25 | 94.68 | 52.29 | 95.49 | 50.28 |
| EBO | 92.13 | 59.70 | 92.06 | 59.69 | 94.34 | 51.74 | 94.44 | 51.62 | 89.88 | 64.17 | 88.64 | 69.83 | 93.11 | 55.97 | 91.01 | 64.62 | 91.76 | 60.07 | 92.41 | 58.27 |
| GEN | 82.48 | 62.53 | 82.28 | 62.57 | 93.63 | 52.45 | 93.48 | 52.93 | 77.89 | 67.02 | 75.82 | 74.07 | 90.15 | 57.00 | 79.96 | 69.69 | 83.69 | 62.36 | 87.38 | 59.32 |
| GradNorm | 93.31 | 53.92 | 93.33 | 53.90 | 94.36 | 51.14 | 94.48 | 50.89 | 91.52 | 55.10 | 91.49 | 53.25 | 93.38 | 52.99 | 93.09 | 52.17 | 92.70 | 53.84 | 92.99 | 54.07 |
| MLS | 91.79 | 59.50 | 91.80 | 59.51 | 94.48 | 51.96 | 94.55 | 52.01 | 89.68 | 63.83 | 88.05 | 68.36 | 93.07 | 56.09 | 90.32 | 63.88 | 91.54 | 59.89 | 92.28 | 58.15 |
| MSP | 88.18 | 58.86 | 87.75 | 58.88 | 94.17 | 51.93 | 94.07 | 52.20 | 84.17 | 62.58 | 82.06 | 65.61 | 91.32 | 55.68 | 85.31 | 62.13 | 88.32 | 59.10 | 90.44 | 57.37 |
| ODIN | 93.31 | 55.45 | 93.27 | 55.48 | 94.71 | 50.75 | 94.74 | 50.84 | 91.23 | 59.83 | 91.33 | 63.56 | 93.96 | 53.20 | 92.01 | 60.30 | 93.27 | 55.60 | 93.61 | 54.34 |
| ReAct | 88.93 | 58.10 | 88.59 | 58.09 | 94.54 | 50.84 | 94.65 | 50.94 | 85.55 | 62.08 | 81.72 | 70.57 | 93.20 | 54.12 | 85.99 | 64.80 | 88.93 | 58.35 | 91.03 | 56.03 |
| CADRef | 89.90 | 56.41 | 89.76 | 56.40 | 94.55 | 51.23 | 94.25 | 51.93 | 88.07 | 60.19 | 80.94 | 65.05 | 93.20 | 54.03 | 85.57 | 60.94 | 90.22 | 56.88 | 92.15 | 55.06 |
| DICE | 86.57 | 60.00 | 86.52 | 59.99 | 94.54 | 50.64 | 93.88 | 51.45 | 81.09 | 64.80 | 80.53 | 70.32 | 91.28 | 55.11 | 84.70 | 64.60 | 87.18 | 59.58 | 89.30 | 57.02 |
| KNN | 91.56 | 54.54 | 91.50 | 54.53 | 94.91 | 50.72 | 94.47 | 51.49 | 90.47 | 57.56 | 84.42 | 61.22 | 93.77 | 52.93 | 87.62 | 58.07 | 91.82 | 54.92 | 93.15 | 53.60 |
| NCI | 92.70 | 53.74 | 92.63 | 53.72 | 95.04 | 50.25 | 95.15 | 50.52 | 92.00 | 56.60 | 84.25 | 64.75 | 94.97 | 51.61 | 83.43 | 61.01 | 91.78 | 54.12 | 94.40 | 52.32 |
| PRO | 81.70 | 60.14 | 81.72 | 60.11 | 92.54 | 53.39 | 92.72 | 53.65 | 80.15 | 62.96 | 74.24 | 65.56 | 87.84 | 57.57 | 83.38 | 61.13 | 82.96 | 60.66 | 85.02 | 59.19 |
| SCALE | 92.21 | 58.91 | 92.16 | 58.88 | 94.08 | 52.44 | 94.22 | 52.30 | 89.82 | 63.04 | 88.84 | 68.25 | 93.02 | 56.30 | 90.94 | 63.43 | 91.78 | 59.20 | 92.20 | 58.23 |
| ViM | 94.11 | 54.34 | 94.15 | 54.32 | 95.56 | 50.25 | 95.29 | 50.53 | 93.18 | 57.02 | 85.72 | 67.94 | 95.13 | 51.49 | 91.48 | 59.74 | 94.05 | 54.49 | 94.83 | 52.28 |
| TriScore | 82.07 | 64.74 | 81.64 | 64.77 | 92.09 | 54.00 | 92.44 | 54.03 | 77.72 | 67.66 | 75.92 | 72.22 | 87.69 | 59.72 | 84.20 | 66.65 | 82.23 | 65.07 | 84.99 | 62.29 |

Table 16: Adversarial OOD detection results on ImageNet with Swin-T.

| Method | AA FPR@95 | AA AUROC | APGD-CE FPR@95 | APGD-CE AUROC | CW FPR@95 | CW AUROC | DF FPR@95 | DF AUROC | FGSM FPR@95 | FGSM AUROC | MPGD FPR@95 | MPGD AUROC | PGD FPR@95 | PGD AUROC | SQUARE FPR@95 | SQUARE AUROC | BIM FPR@95 | BIM AUROC | PGD_BPDA FPR@95 | PGD_BPDA AUROC |
|---|---|---|---|---|---|---|---|---|---|---|---|---|---|---|---|---|---|---|---|---|
| ASH | 93.86 | 53.92 | 93.98 | 54.15 | 94.57 | 51.89 | 94.71 | 51.66 | 93.15 | 54.20 | 92.16 | 51.58 | 93.95 | 53.99 | 97.41 | 40.07 | 93.30 | 54.93 | 93.82 | 54.76 |
| EBO | 91.90 | 58.76 | 92.01 | 58.80 | 93.30 | 55.33 | 93.32 | 55.15 | 90.20 | 62.79 | 89.70 | 63.45 | 92.13 | 58.62 | 91.26 | 60.27 | 90.85 | 60.86 | 91.40 | 59.99 |
| GEN | 76.86 | 67.92 | 76.24 | 68.00 | 91.28 | 55.84 | 91.20 | 56.20 | 75.03 | 70.53 | 70.65 | 72.78 | 86.67 | 61.69 | 81.16 | 67.21 | 76.25 | 68.79 | 81.13 | 65.33 |
| GradNorm | 94.25 | 45.62 | 94.24 | 45.62 | 93.39 | 53.01 | 93.68 | 52.10 | 93.00 | 47.71 | 92.90 | 44.71 | 93.72 | 51.32 | 93.31 | 48.51 | 93.72 | 46.70 | 93.59 | 50.17 |
| MLS | 91.63 | 59.57 | 91.68 | 59.60 | 93.34 | 55.09 | 93.32 | 55.03 | 90.02 | 63.57 | 89.46 | 63.24 | 91.96 | 58.78 | 91.01 | 60.18 | 90.45 | 61.58 | 91.22 | 60.28 |
| MSP | 88.02 | 61.01 | 87.86 | 61.06 | 92.87 | 54.50 | 92.80 | 54.63 | 86.48 | 64.11 | 84.65 | 63.07 | 90.36 | 58.62 | 87.88 | 59.99 | 86.34 | 62.47 | 88.64 | 60.43 |
| ODIN | 93.85 | 53.14 | 93.82 | 53.22 | 94.86 | 50.99 | 94.86 | 50.80 | 91.13 | 59.00 | 93.47 | 56.14 | 93.85 | 53.31 | 93.39 | 54.53 | 93.01 | 54.41 | 93.47 | 53.93 |
| ReAct | 86.06 | 63.25 | 85.91 | 63.32 | 92.90 | 54.29 | 92.73 | 54.67 | 84.86 | 65.79 | 80.80 | 68.68 | 89.97 | 59.23 | 87.15 | 62.73 | 84.47 | 64.68 | 87.55 | 61.72 |
| CADRef | 78.42 | 65.87 | 78.10 | 65.90 | 92.53 | 53.00 | 92.36 | 53.74 | 76.86 | 67.89 | 68.47 | 69.77 | 88.63 | 58.58 | 84.41 | 63.58 | 78.92 | 66.22 | 84.28 | 61.81 |
| DICE | 86.76 | 69.89 | 86.67 | 69.87 | 93.02 | 54.25 | 93.13 | 54.49 | 79.14 | 74.03 | 93.94 | 61.36 | 89.73 | 60.88 | 86.93 | 72.54 | 84.44 | 70.68 | 87.82 | 64.45 |
| KNN | 83.74 | 61.83 | 83.57 | 61.85 | 93.47 | 51.51 | 93.35 | 52.26 | 83.20 | 63.49 | 76.54 | 65.13 | 91.06 | 55.74 | 86.65 | 60.41 | 84.20 | 62.02 | 88.05 | 58.20 |
| NCI | 78.82 | 64.89 | 79.01 | 64.92 | 93.29 | 52.63 | 93.06 | 53.26 | 80.43 | 66.42 | 72.69 | 68.84 | 89.83 | 58.01 | 84.61 | 63.46 | 80.54 | 65.37 | 85.91 | 61.16 |
| PRO | 76.77 | 65.45 | 76.93 | 65.49 | 88.48 | 57.53 | 88.40 | 57.82 | 79.80 | 65.83 | 75.61 | 65.41 | 84.45 | 61.50 | 81.94 | 62.65 | 77.17 | 66.37 | 80.24 | 64.02 |
| SCALE | 92.24 | 56.26 | 92.23 | 56.33 | 93.24 | 55.00 | 93.33 | 54.65 | 90.65 | 59.72 | 92.13 | 55.27 | 92.32 | 57.10 | 91.37 | 57.60 | 91.69 | 58.09 | | |
| ViM | 83.60 | 67.87 | 83.52 | 67.91 | 94.45 | 51.75 | 94.03 | 52.60 | 79.63 | 68.53 | 71.04 | 73.99 | 90.37 | 58.41 | 88.47 | 62.74 | 81.13 | 68.13 | 86.42 | 62.75 |
| TriScore | 74.60 | 69.83 | 74.60 | 69.85 | 89.60 | 58.01 | 89.60 | 58.28 | 76.72 | 68.70 | 73.77 | 68.46 | 84.50 | 62.87 | 82.98 | 65.19 | 74.59 | 70.70 | 79.11 | 67.20 |

## C.5 Clean OOD detection branch-isolation by backbone and dataset

Table 17: Clean OOD branch-isolation results on ImageNet with ResNet-50.

| Family | Score | Near-OOD ninco FPR@95 | Near-OOD ninco AUROC | Near-OOD ssb_hard FPR@95 | Near-OOD ssb_hard AUROC | Far-OOD inaturalist FPR@95 | Far-OOD inaturalist AUROC | Far-OOD openimage_o FPR@95 | Far-OOD openimage_o AUROC | Far-OOD textures FPR@95 | Far-OOD textures AUROC |
|---|---|---|---|---|---|---|---|---|---|---|---|
| Energy | EBO | 60.56 | 79.70 | 76.52 | 72.08 | 31.30 | 90.63 | 38.07 | 89.06 | 45.81 | 88.70 |
| Energy | E (Ours) | 62.45 | 77.36 | 78.90 | 68.10 | 34.68 | 87.87 | 38.27 | 87.83 | 42.81 | 88.78 |
| Boundary | Margin-only | 52.92 | 81.13 | 73.27 | 73.17 | 39.39 | 87.63 | 45.75 | 84.89 | 57.99 | 82.15 |
| Boundary | GradNorm | 79.50 | 74.01 | 78.22 | 71.90 | 32.08 | 93.89 | 68.51 | 84.82 | 43.36 | 92.04 |
| Boundary | **BP (Ours)** | 51.17 | 81.75 | 74.43 | 72.56 | 37.36 | 88.04 | 42.13 | 85.93 | 51.64 | 84.12 |
| Consistency | Argmax | 100.00 | 64.23 | 100.00 | 57.80 | 100.00 | 68.88 | 100.00 | 63.41 | 100.00 | 63.21 |
| Consistency | TTA entropy | 63.26 | 73.72 | 77.94 | 64.94 | 55.08 | 79.54 | 60.30 | 73.90 | 65.56 | 73.40 |
| Consistency | **TC (Ours)** | 56.06 | 76.74 | 77.68 | 66.36 | 41.55 | 80.88 | 52.39 | 74.70 | 61.03 | 72.42 |

Table 18: Clean OOD branch-isolation results on ImageNet with RegNet.

| Family | Score | Near-OOD ninco FPR@95 | Near-OOD ninco AUROC | Near-OOD ssb_hard FPR@95 | Near-OOD ssb_hard AUROC | Far-OOD inaturalist FPR@95 | Far-OOD inaturalist AUROC | Far-OOD openimage_o FPR@95 | Far-OOD openimage_o AUROC | Far-OOD textures FPR@95 | Far-OOD textures AUROC |
|---|---|---|---|---|---|---|---|---|---|---|---|
| Energy | EBO | 42.53 | 91.67 | 62.01 | 85.29 | 7.72 | 98.29 | 26.03 | 95.83 | 38.11 | 93.02 |
| Energy | E (Ours) | 42.95 | 91.25 | 65.25 | 83.53 | 8.72 | 98.08 | 22.39 | 96.18 | 36.57 | 93.68 |
| Boundary | Margin-only | 47.94 | 87.43 | 67.92 | 78.77 | 26.84 | 93.36 | 34.22 | 91.29 | 44.66 | 88.58 |
| Boundary | GradNorm | 93.25 | 49.01 | 91.59 | 49.42 | 88.22 | 56.99 | 92.57 | 60.44 | 71.55 | 76.12 |
| Boundary | **BP (Ours)** | 46.84 | 87.57 | 68.89 | 78.47 | 26.41 | 93.47 | 33.16 | 91.61 | 42.14 | 89.10 |
| Consistency | Argmax | 100.00 | 60.57 | 100.00 | 55.93 | 100.00 | 66.38 | 100.00 | 62.20 | 100.00 | 64.28 |
| Consistency | TTA entropy | 59.19 | 81.20 | 77.91 | 72.13 | 49.46 | 84.91 | 58.79 | 81.49 | 57.72 | 83.17 |
| Consistency | **TC (Ours)** | 48.71 | 83.40 | 75.17 | 73.07 | 25.22 | 89.53 | 37.06 | 84.96 | 44.73 | 86.21 |

Table 19: Clean OOD branch-isolation results on ImageNet with ViT-B-16.

| Family | Score | Near-OOD ninco FPR@95 | Near-OOD ninco AUROC | Near-OOD ssb_hard FPR@95 | Near-OOD ssb_hard AUROC | Far-OOD inaturalist FPR@95 | Far-OOD inaturalist AUROC | Far-OOD openimage_o FPR@95 | Far-OOD openimage_o AUROC | Far-OOD textures FPR@95 | Far-OOD textures AUROC |
|---|---|---|---|---|---|---|---|---|---|---|---|
| Energy | EBO | 94.18 | 66.02 | 92.22 | 58.80 | 83.64 | 79.30 | 88.82 | 76.48 | 83.62 | 81.16 |
| Energy | E (Ours) | 94.22 | 66.32 | 92.44 | 58.61 | 84.43 | 78.80 | 89.30 | 76.13 | 84.04 | 81.09 |
| Boundary | Margin-only | 56.21 | 81.73 | 80.51 | 71.71 | 36.66 | 88.94 | 43.16 | 86.82 | 47.94 | 85.96 |
| Boundary | GradNorm | 95.79 | 35.60 | 93.62 | 42.96 | 91.17 | 42.42 | 94.51 | 37.83 | 92.27 | 45.00 |
| Boundary | **BP (Ours)** | 55.05 | 82.21 | 84.06 | 70.18 | 32.82 | 89.80 | 37.28 | 88.20 | 39.98 | 87.98 |
| Consistency | Argmax | 100.00 | 63.69 | 100.00 | 57.13 | 100.00 | 69.96 | 100.00 | 65.58 | 100.00 | 65.16 |
| Consistency | TTA entropy | 77.90 | 72.53 | 90.67 | 61.09 | 68.24 | 77.40 | 74.32 | 74.22 | 78.69 | 72.06 |
| Consistency | **TC (Ours)** | 57.34 | 80.86 | 85.34 | 67.90 | 33.48 | 87.45 | 43.76 | 83.26 | 51.77 | 81.28 |

Table 20: Clean OOD branch-isolation results on ImageNet with Swin-T.

| Family | Score | Near-OOD ninco FPR@95 | Near-OOD ninco AUROC | Near-OOD ssb_hard FPR@95 | Near-OOD ssb_hard AUROC | Far-OOD inaturalist FPR@95 | Far-OOD inaturalist AUROC | Far-OOD openimage_o FPR@95 | Far-OOD openimage_o AUROC | Far-OOD textures FPR@95 | Far-OOD textures AUROC |
|---|---|---|---|---|---|---|---|---|---|---|---|
| Energy | EBO | 79.16 | 78.25 | 87.45 | 68.21 | 61.52 | 85.13 | 80.81 | 79.86 | 84.45 | 78.96 |
| Energy | E (Ours) | 77.98 | 78.85 | 87.09 | 68.46 | 61.04 | 85.30 | 80.04 | 80.08 | 83.66 | 79.00 |
| Boundary | Margin-only | 53.15 | 82.92 | 76.26 | 73.17 | 34.94 | 89.42 | 44.08 | 86.53 | 54.91 | 83.60 |
| Boundary | GradNorm | 94.07 | 44.81 | 93.57 | 50.36 | 95.12 | 38.23 | 96.99 | 33.47 | 97.73 | 34.71 |
| Boundary | **BP (Ours)** | 51.10 | 83.35 | 80.31 | 72.19 | 32.31 | 90.01 | 38.74 | 87.81 | 47.37 | 85.60 |
| Consistency | Argmax | 100.00 | 63.36 | 100.00 | 56.37 | 100.00 | 69.78 | 100.00 | 63.47 | 100.00 | 64.23 |
| Consistency | TTA entropy | 83.65 | 66.65 | 91.63 | 56.78 | 82.50 | 68.40 | 86.56 | 64.46 | 87.27 | 65.69 |
| Consistency | **TC (Ours)** | 58.62 | 78.40 | 87.36 | 64.03 | 36.05 | 85.76 | 52.19 | 77.99 | 61.46 | 78.06 |

## C.6 Adversarial OOD detection branch-isolation by backbone and attack

Table 21: Adversarial OOD branch-isolation results on ImageNet with ResNet-50.

| Family | Score | AA FPR@95 | AA AUROC | APGD-CE FPR@95 | APGD-CE AUROC | C&W FPR@95 | C&W AUROC | DF FPR@95 | DF AUROC | FGSM FPR@95 | FGSM AUROC | MPGD FPR@95 | MPGD AUROC | PGD FPR@95 | PGD AUROC | SQUARE FPR@95 | SQUARE AUROC | BIM FPR@95 | BIM AUROC | PGD_BPDA FPR@95 | PGD_BPDA AUROC |
|---|---|---|---|---|---|---|---|---|---|---|---|---|---|---|---|---|---|---|---|---|---|
| Energy | EBO | 96.22 | 49.83 | 96.34 | 49.85 | 89.78 | 59.55 | 88.23 | 60.95 | 78.17 | 73.49 | 92.23 | 52.61 | 93.95 | 49.37 | 88.78 | 62.33 | 96.86 | 44.32 | 96.07 | 43.71 |
| Energy | E (Ours) | 96.19 | 49.99 | 96.25 | 50.02 | 89.80 | 58.79 | 88.32 | 60.30 | 77.68 | 72.52 | 91.83 | 52.81 | 93.83 | 49.65 | 88.78 | 61.55 | 96.82 | 44.98 | 96.06 | 44.30 |
| Boundary | Margin-only | 89.82 | 57.27 | 89.90 | 57.26 | 88.25 | 59.41 | 87.25 | 60.35 | 74.34 | 70.47 | 87.62 | 55.56 | 86.01 | 55.01 | 86.57 | 63.08 | 91.23 | 50.39 | 89.29 | 50.88 |
| Boundary | GradNorm | 98.18 | 43.96 | 98.19 | 43.94 | 88.79 | 62.20 | 88.22 | 63.44 | 81.21 | 72.42 | 97.72 | 39.93 | 96.04 | 47.21 | 89.47 | 61.85 | 98.11 | 39.40 | 97.75 | 40.75 |
| Boundary | **BP (Ours)** | 84.79 | 61.86 | 84.74 | 61.85 | 87.84 | 58.08 | 88.52 | 57.94 | 74.35 | 70.38 | 86.49 | 56.31 | 85.99 | 55.72 | 86.64 | 63.23 | 91.82 | 50.99 | 89.37 | 51.65 |
| Consistency | Argmax | 100.00 | 59.80 | 100.00 | 59.78 | 100.00 | 52.99 | 100.00 | 53.54 | 100.00 | 57.90 | 100.00 | 56.13 | 100.00 | 57.20 | 100.00 | 54.54 | 100.00 | 56.91 | 100.00 | 57.13 |
| Consistency | TTA entropy | 81.71 | 66.33 | 81.78 | 66.40 | 89.24 | 56.58 | 88.72 | 57.45 | 75.18 | 67.96 | 82.29 | 64.78 | 79.00 | 67.05 | 86.76 | 60.80 | 78.47 | 68.43 | 79.00 | 67.59 |
| Consistency | **TC (Ours)** | 80.54 | 69.98 | 80.57 | 70.02 | 89.30 | 56.67 | 88.78 | 57.61 | 74.71 | 67.80 | 81.60 | 66.44 | 78.08 | 68.79 | 86.92 | 61.40 | 76.56 | 69.92 | 77.60 | 69.56 |

Table 22: Adversarial OOD branch-isolation results on ImageNet with RegNet.

| Family | Score | AA FPR@95 | AA AUROC | APGD-CE FPR@95 | APGD-CE AUROC | C&W FPR@95 | C&W AUROC | DF FPR@95 | DF AUROC | FGSM FPR@95 | FGSM AUROC | MPGD FPR@95 | MPGD AUROC | PGD FPR@95 | PGD AUROC | SQUARE FPR@95 | SQUARE AUROC | BIM FPR@95 | BIM AUROC | PGD_BPDA FPR@95 | PGD_BPDA AUROC |
|---|---|---|---|---|---|---|---|---|---|---|---|---|---|---|---|---|---|---|---|---|---|
| Energy | EBO | 80.82 | 71.83 | 80.51 | 71.85 | 87.02 | 66.25 | 87.17 | 66.17 | 79.08 | 73.27 | 83.11 | 71.50 | 84.05 | 68.26 | 83.47 | 70.84 | 80.69 | 72.28 | 81.69 | 70.69 |
| Energy | E (Ours) | 81.07 | 71.34 | 81.27 | 71.36 | 87.56 | 65.61 | 87.63 | 65.76 | 79.40 | 72.74 | 83.34 | 71.13 | 84.57 | 67.85 | 83.85 | 70.41 | 81.07 | 71.78 | 82.43 | 70.22 |
| Boundary | Margin-only | 85.44 | 61.71 | 85.42 | 61.69 | 88.25 | 58.08 | 88.86 | 57.87 | 84.67 | 64.34 | 85.23 | 61.78 | 87.86 | 59.85 | 85.54 | 61.12 | 85.40 | 62.75 | 86.53 | 61.17 |
| Boundary | GradNorm | 96.38 | 43.47 | 96.36 | 43.42 | 91.09 | 55.95 | 91.14 | 56.31 | 94.50 | 47.01 | 97.08 | 35.44 | 93.80 | 50.83 | 94.29 | 47.88 | 96.54 | 42.78 | 95.14 | 47.36 |
| Boundary | **BP (Ours)** | 84.79 | 61.86 | 84.74 | 61.85 | 87.84 | 58.08 | 88.52 | 57.94 | 74.35 | 70.38 | 86.49 | 56.31 | 87.52 | 59.93 | 84.85 | 61.36 | 84.70 | 62.90 | 85.96 | 61.28 |
| Consistency | Argmax | 100.00 | 51.44 | 100.00 | 51.37 | 100.00 | 50.62 | 100.00 | 50.58 | 100.00 | 52.80 | 100.00 | 51.61 | 100.00 | 51.63 | 100.00 | 51.31 | 100.00 | 50.89 | 100.00 | 51.68 |
| Consistency | TTA entropy | 87.96 | 61.86 | 87.57 | 62.08 | 92.52 | 55.59 | 92.85 | 55.57 | 88.07 | 62.94 | 87.87 | 63.04 | 90.78 | 58.59 | 89.51 | 59.97 | 87.78 | 63.05 | 89.07 | 60.76 |
| Consistency | **TC (Ours)** | 84.16 | 62.42 | 84.05 | 62.46 | 91.67 | 54.88 | 92.19 | 54.85 | 86.58 | 63.18 | 83.38 | 63.83 | 89.38 | 58.82 | 87.85 | 59.65 | 84.28 | 63.77 | 86.85 | 61.20 |

Table 23: Adversarial OOD branch-isolation results on ImageNet with ViT-B-16.

| Family | Score | AA FPR@95 | AA AUROC | APGD-CE FPR@95 | APGD-CE AUROC | C&W FPR@95 | C&W AUROC | DF FPR@95 | DF AUROC | FGSM FPR@95 | FGSM AUROC | MPGD FPR@95 | MPGD AUROC | PGD FPR@95 | PGD AUROC | SQUARE FPR@95 | SQUARE AUROC | BIM FPR@95 | BIM AUROC | PGD_BPDA FPR@95 | PGD_BPDA AUROC |
|---|---|---|---|---|---|---|---|---|---|---|---|---|---|---|---|---|---|---|---|---|---|
| Energy | EBO | 92.11 | 59.70 | 92.03 | 59.69 | 94.34 | 51.74 | 94.42 | 51.62 | 89.89 | 64.17 | 88.64 | 69.83 | 93.10 | 55.97 | 90.98 | 64.62 | 91.76 | 60.07 | 92.41 | 58.27 |
| Energy | E (Ours) | 92.05 | 59.62 | 92.03 | 59.61 | 94.41 | 51.86 | 94.52 | 51.62 | 89.83 | 64.04 | 88.64 | 69.70 | 93.08 | 55.92 | 90.98 | 64.53 | 91.24 | 60.11 | 90.83 | 60.70 |
| Boundary | Margin-only | 88.99 | 56.52 | 88.74 | 56.50 | 94.07 | 52.13 | 94.12 | 52.49 | 86.84 | 60.16 | 82.32 | 62.49 | 92.32 | 54.94 | 85.03 | 59.52 | 89.39 | 57.19 | 91.02 | 55.90 |
| Boundary | GradNorm | 93.31 | 53.92 | 93.32 | 53.90 | 94.39 | 51.14 | 94.47 | 50.89 | 91.53 | 55.09 | 97.08 | 35.44 | 93.72 | 51.32 | 93.31 | 48.51 | 93.72 | 46.70 | 93.59 | 50.17 |
| Boundary | **BP (Ours)** | 88.73 | 57.08 | 88.66 | 57.07 | 94.54 | 51.69 | 94.20 | 52.37 | 86.36 | 60.70 | 80.37 | 64.89 | 92.57 | 54.85 | 83.41 | 61.29 | 89.27 | 57.60 | 91.25 | 55.95 |
| Consistency | Argmax | 100.00 | 51.32 | 100.00 | 51.31 | 100.00 | 50.58 | 100.00 | 50.65 | 100.00 | 53.23 | 100.00 | 51.13 | 100.00 | 51.26 | 100.00 | 51.13 | 100.00 | 51.76 | 100.00 | 51.49 |
| Consistency | TTA entropy | 90.81 | 55.97 | 90.90 | 55.88 | 94.92 | 50.57 | 94.91 | 50.90 | 89.67 | 58.38 | 82.15 | 66.21 | 93.19 | 53.64 | 90.93 | 56.72 | 91.16 | 56.26 | 91.87 | 55.20 |
| Consistency | **TC (Ours)** | 82.82 | 59.07 | 82.78 | 59.08 | 94.48 | 51.58 | 94.52 | 51.95 | 80.58 | 62.38 | 66.42 | 70.67 | 91.26 | 55.32 | 81.60 | 62.31 | 84.25 | 59.22 | 88.16 | 57.16 |

Table 24: Adversarial OOD branch-isolation results on ImageNet with Swin-T.

| Family | Score | AA FPR@95 | AA AUROC | APGD-CE FPR@95 | APGD-CE AUROC | C&W FPR@95 | C&W AUROC | DF FPR@95 | DF AUROC | FGSM FPR@95 | FGSM AUROC | MPGD FPR@95 | MPGD AUROC | PGD FPR@95 | PGD AUROC | SQUARE FPR@95 | SQUARE AUROC | BIM FPR@95 | BIM AUROC | PGD_BPDA FPR@95 | PGD_BPDA AUROC |
|---|---|---|---|---|---|---|---|---|---|---|---|---|---|---|---|---|---|---|---|---|---|
| Energy | EBO | 91.90 | 58.76 | 92.01 | 58.80 | 93.30 | 55.33 | 93.32 | 55.15 | 90.20 | 62.79 | 89.70 | 63.45 | 92.13 | 58.62 | 91.26 | 60.27 | 90.85 | 60.86 | 91.40 | 59.99 |
| Energy | E (Ours) | 91.92 | 58.63 | 91.97 | 58.67 | 93.31 | 55.36 | 93.34 | 55.09 | 90.29 | 62.62 | 89.82 | 63.27 | 92.07 | 58.52 | 91.24 | 60.11 | 90.83 | 60.70 | 91.40 | 59.86 |
| Boundary | Margin-only | 82.40 | 61.36 | 82.31 | 61.40 | 92.07 | 53.74 | 92.24 | 53.94 | 82.51 | 64.02 | 78.56 | 63.04 | 89.57 | 57.60 | 85.66 | 59.59 | 82.86 | 62.28 | 86.63 | 59.45 |
| Boundary | GradNorm | 94.25 | 45.62 | 94.32 | 45.62 | 93.39 | 53.01 | 93.68 | 52.10 | 93.00 | 47.71 | 92.90 | 44.71 | 93.72 | 51.32 | 93.31 | 48.51 | 93.72 | 46.70 | 93.59 | 50.17 |
| Boundary | **BP (Ours)** | 79.77 | 63.15 | 79.52 | 63.20 | 91.62 | 53.89 | 91.81 | 54.24 | 78.74 | 65.51 | 73.94 | 65.54 | 88.15 | 58.31 | 83.88 | 61.24 | 80.05 | 63.83 | 84.71 | 60.57 |
| Consistency | Argmax | 100.00 | 51.67 | 100.00 | 51.64 | 100.00 | 50.08 | 100.00 | 50.10 | 100.00 | 53.23 | 100.00 | 51.13 | 100.00 | 51.43 | 100.00 | 51.13 | 100.00 | 51.76 | 100.00 | 51.82 |
| Consistency | TTA entropy | 92.08 | 54.79 | 91.95 | 54.87 | 94.78 | 49.97 | 94.58 | 50.19 | 91.82 | 55.72 | 89.35 | 58.73 | 93.92 | 51.64 | 92.37 | 54.55 | 92.21 | 55.12 | 92.76 | 53.64 |
| Consistency | **TC (Ours)** | 78.30 | 62.29 | 78.19 | 62.34 | 93.38 | 50.39 | 93.42 | 50.69 | 79.42 | 63.46 | 69.26 | 67.37 | 89.46 | 55.75 | 84.43 | 59.82 | 79.39 | 62.81 | 84.04 | 59.41 |

### C.7 Clean OOD detection branch-combination by backbone and dataset

Table 25: Clean OOD branch-combination results on ImageNet with ResNet-50.

| Method | Near-OOD ninco FPR@95 | Near-OOD ninco AUROC | Near-OOD ssb_hard FPR@95 | Near-OOD ssb_hard AUROC | Far-OOD inaturalist FPR@95 | Far-OOD inaturalist AUROC | Far-OOD openimage_o FPR@95 | Far-OOD openimage_o AUROC | Far-OOD textures FPR@95 | Far-OOD textures AUROC |
|---|---|---|---|---|---|---|---|---|---|---|
| E | 59.83 | 80.29 | 76.27 | 72.42 | 30.51 | 91.14 | 37.77 | 89.19 | 46.25 | 88.49 |
| E+TC | 57.28 | 80.38 | 75.95 | 71.98 | 31.68 | 87.16 | 37.99 | 84.97 | 46.50 | 83.40 |
| E+BP | 53.97 | 80.80 | 73.45 | 73.05 | 40.26 | 85.56 | 43.51 | 84.84 | 60.38 | 80.26 |
| BP | 60.32 | 77.23 | 77.69 | 70.24 | 63.22 | 76.27 | 65.43 | 76.70 | 88.86 | 68.20 |
| BP+TC | 60.14 | 78.00 | 77.48 | 70.22 | 61.10 | 78.82 | 64.16 | 77.15 | 88.39 | 70.35 |
| TC | 55.00 | 78.47 | 75.12 | 69.58 | 41.68 | 80.41 | 49.69 | 77.11 | 61.97 | 72.85 |
| E+BP+TC | 53.69 | 80.30 | 76.16 | 69.91 | 31.93 | 87.51 | 38.61 | 84.12 | 47.47 | 83.13 |
| TriScore | 53.70 | 80.17 | 76.16 | 69.91 | 31.93 | 87.47 | 38.64 | 84.22 | 47.48 | 83.28 |

Table 26: Clean OOD branch-combination results on ImageNet with RegNet.

| Method | Near-OOD ninco FPR@95 | Near-OOD ninco AUROC | Near-OOD ssb_hard FPR@95 | Near-OOD ssb_hard AUROC | Far-OOD inaturalist FPR@95 | Far-OOD inaturalist AUROC | Far-OOD openimage_o FPR@95 | Far-OOD openimage_o AUROC | Far-OOD textures FPR@95 | Far-OOD textures AUROC |
|---|---|---|---|---|---|---|---|---|---|---|
| E | 42.55 | 91.67 | 62.09 | 85.28 | 7.72 | 98.29 | 25.95 | 95.83 | 38.18 | 93.02 |
| E+TC | 39.01 | 90.36 | 62.04 | 83.04 | 11.43 | 95.97 | 22.76 | 93.95 | 37.98 | 91.24 |
| E+BP | 38.26 | 90.31 | 57.99 | 83.21 | 18.23 | 95.94 | 25.38 | 94.32 | 40.61 | 90.38 |
| BP | 54.59 | 82.11 | 72.73 | 74.86 | 49.39 | 83.90 | 48.42 | 84.43 | 65.60 | 80.39 |
| BP+TC | 54.54 | 82.56 | 72.76 | 74.87 | 47.39 | 85.42 | 47.67 | 84.76 | 65.35 | 81.65 |
| TC | 48.79 | 83.53 | 71.03 | 75.30 | 27.51 | 87.52 | 37.38 | 84.69 | 47.17 | 84.40 |
| E+BP+TC | 40.25 | 89.77 | 64.34 | 80.94 | 14.62 | 95.69 | 23.54 | 93.68 | 33.12 | 92.02 |
| TriScore | 40.28 | 89.42 | 64.36 | 80.87 | 13.20 | 95.15 | 23.62 | 93.50 | 33.31 | 91.58 |

Table 27: Clean OOD branch-combination results on ImageNet with ViT-B-16.

| Method | Near-OOD ninco FPR@95 | Near-OOD ninco AUROC | Near-OOD ssb_hard FPR@95 | Near-OOD ssb_hard AUROC | Far-OOD inaturalist FPR@95 | Far-OOD inaturalist AUROC | Far-OOD openimage_o FPR@95 | Far-OOD openimage_o AUROC | Far-OOD textures FPR@95 | Far-OOD textures AUROC |
|---|---|---|---|---|---|---|---|---|---|---|
| E | 94.16 | 66.02 | 92.25 | 58.80 | 83.58 | 79.30 | 88.79 | 76.48 | 83.65 | 81.17 |
| E+TC | 92.75 | 74.06 | 91.38 | 64.90 | 70.60 | 85.60 | 85.33 | 81.59 | 78.72 | 83.09 |
| E+BP | 79.72 | 76.48 | 86.85 | 67.15 | 57.11 | 84.18 | 68.41 | 82.27 | 69.12 | 82.69 |
| BP | 57.88 | 79.59 | 82.42 | 68.94 | 52.28 | 81.40 | 53.77 | 80.93 | 72.90 | 76.66 |
| BP+TC | 55.94 | 81.52 | 82.39 | 69.18 | 45.92 | 85.36 | 49.63 | 82.67 | 67.14 | 79.05 |
| TC | 57.76 | 81.66 | 86.05 | 67.92 | 35.13 | 87.44 | 45.46 | 83.21 | 53.67 | 81.07 |
| E+BP+TC | 78.42 | 79.29 | 88.56 | 67.74 | 37.88 | 89.17 | 55.65 | 85.68 | 53.01 | 86.54 |
| TriScore | 77.99 | 79.07 | 88.55 | 67.66 | 37.82 | 88.74 | 55.24 | 85.25 | 53.05 | 85.93 |

Table 28: Clean OOD branch-combination results on ImageNet with Swin-T.

| Method | Near-OOD ninco FPR@95 | Near-OOD ninco AUROC | Near-OOD ssb_hard FPR@95 | Near-OOD ssb_hard AUROC | Far-OOD inaturalist FPR@95 | Far-OOD inaturalist AUROC | Far-OOD openimage_o FPR@95 | Far-OOD openimage_o AUROC | Far-OOD textures FPR@95 | Far-OOD textures AUROC |
|---|---|---|---|---|---|---|---|---|---|---|
| E | 79.16 | 78.25 | 87.45 | 68.21 | 61.52 | 85.13 | 80.81 | 79.86 | 84.45 | 78.96 |
| E+TC | 77.44 | 80.30 | 87.16 | 69.55 | 54.52 | 87.00 | 78.30 | 81.40 | 82.39 | 79.74 |
| E+BP | 63.14 | 82.01 | 83.56 | 71.21 | 45.05 | 86.07 | 60.94 | 82.95 | 78.51 | 78.62 |
| BP | 60.92 | 79.45 | 82.87 | 69.04 | 54.24 | 80.63 | 57.85 | 79.82 | 83.24 | 72.41 |
| BP+TC | 57.45 | 81.53 | 82.78 | 69.74 | 45.14 | 84.75 | 52.14 | 81.55 | 79.50 | 75.46 |
| TC | 55.08 | 81.71 | 83.80 | 69.14 | 35.23 | 86.60 | 47.15 | 81.84 | 61.65 | 78.42 |
| E+BP+TC | 56.94 | 83.01 | 83.21 | 70.70 | 29.48 | 90.42 | 47.39 | 85.23 | 59.66 | 84.26 |
| TriScore | 56.98 | 82.47 | 83.21 | 70.51 | 29.52 | 89.81 | 47.46 | 84.97 | 59.63 | 83.95 |

## C.8 Adversarial OOD detection branch-combination by backbone and attack

Table 29: Adversarial OOD branch-combination results on ImageNet with ResNet-50.

| Method | AA | | APGD-CE | | CW | | DF | | FGSM | | MPGD | | PGD | | SQUARE | | BIM | | PGD_BPDA | |
|---|---|---|---|---|---|---|---|---|---|---|---|---|---|---|---|---|---|---|---|---|
| | FPR@95 | AUROC | FPR@95 | AUROC | FPR@95 | AUROC | FPR@95 | AUROC | FPR@95 | AUROC | FPR@95 | AUROC | FPR@95 | AUROC | FPR@95 | AUROC | FPR@95 | AUROC | FPR@95 | AUROC |
| E | 96.17 | 50.19 | 96.29 | 50.20 | 89.62 | 59.62 | 88.14 | 61.00 | 77.76 | 73.43 | 92.12 | 52.70 | 93.83 | 49.51 | 88.62 | 62.51 | 96.83 | 44.40 | 96.03 | 43.83 |
| E+TC | 93.19 | 59.92 | 93.22 | 59.97 | 89.56 | 57.93 | 88.16 | 58.98 | 76.00 | 69.79 | 89.32 | 60.58 | 91.24 | 59.10 | 88.30 | 60.58 | 94.45 | 58.84 | 93.59 | 57.27 |
| E+BP | 87.86 | 60.62 | 87.86 | 60.64 | 87.48 | 61.11 | 85.87 | 62.19 | 74.34 | 71.50 | 85.11 | 59.71 | 84.94 | 59.24 | 86.34 | 65.21 | 90.54 | 54.68 | 88.26 | 55.68 |
| BP | 75.11 | 69.48 | 74.81 | 69.49 | 87.06 | 60.53 | 87.48 | 61.09 | 81.92 | 66.44 | 79.87 | 64.37 | 76.24 | 66.56 | 86.71 | 64.97 | 74.08 | 65.01 | 74.03 | 66.40 |
| BP+TC | 75.18 | 69.47 | 74.90 | 69.51 | 87.10 | 58.89 | 87.51 | 59.39 | 81.75 | 66.15 | 79.94 | 65.35 | 76.36 | 66.95 | 86.74 | 62.50 | 74.21 | 67.75 | 74.19 | 67.58 |
| TC | 82.85 | 66.39 | 83.08 | 66.45 | 89.86 | 55.51 | 89.69 | 55.98 | 78.82 | 64.14 | 83.56 | 64.61 | 81.91 | 65.48 | 87.90 | 58.10 | 79.72 | 68.83 | 81.40 | 66.93 |
| E+BP+TC | 92.59 | 61.39 | 92.73 | 61.44 | 88.73 | 58.57 | 87.08 | 59.91 | 74.27 | 70.86 | 88.63 | 59.98 | 89.62 | 59.25 | 87.27 | 62.37 | 94.83 | 56.37 | 92.82 | 56.49 |
| TriScore | 92.12 | 63.58 | 92.30 | 63.65 | 89.08 | 58.85 | 87.44 | 60.16 | 74.06 | 71.15 | 88.34 | 61.40 | 89.39 | 61.61 | 87.40 | 62.65 | 94.03 | 58.76 | 92.27 | 59.25 |

Table 30: Adversarial OOD branch-combination results on ImageNet with RegNet.

| Method | AA | | APGD-CE | | CW | | DF | | FGSM | | MPGD | | PGD | | SQUARE | | BIM | | PGD_BPDA | |
|---|---|---|---|---|---|---|---|---|---|---|---|---|---|---|---|---|---|---|---|---|
| | FPR@95 | AUROC | FPR@95 | AUROC | FPR@95 | AUROC | FPR@95 | AUROC | FPR@95 | AUROC | FPR@95 | AUROC | FPR@95 | AUROC | FPR@95 | AUROC | FPR@95 | AUROC | FPR@95 | AUROC |
| E | 80.84 | 71.82 | 80.53 | 71.85 | 86.99 | 66.25 | 87.16 | 66.17 | 79.07 | 73.26 | 83.12 | 71.50 | 84.02 | 68.27 | 83.47 | 70.84 | 80.70 | 72.28 | 81.68 | 70.69 |
| E+TC | 81.11 | 68.40 | 80.96 | 68.38 | 87.26 | 63.30 | 87.44 | 63.22 | 79.69 | 69.80 | 83.52 | 67.96 | 84.30 | 65.30 | 83.68 | 67.28 | 81.23 | 68.86 | 82.22 | 67.42 |
| E+BP | 69.84 | 72.46 | 70.37 | 72.45 | 80.82 | 65.89 | 81.24 | 65.84 | 74.79 | 71.49 | 75.20 | 70.72 | 77.30 | 68.62 | 75.81 | 70.01 | 71.03 | 73.03 | 73.39 | 71.19 |
| BP | 74.48 | 68.01 | 74.22 | 67.99 | 83.30 | 61.44 | 83.73 | 61.48 | 82.46 | 64.97 | 77.14 | 65.28 | 81.58 | 64.43 | 80.80 | 64.34 | 75.08 | 68.82 | 78.06 | 66.89 |
| BP+TC | 75.87 | 65.10 | 75.50 | 65.08 | 84.16 | 59.16 | 84.58 | 59.18 | 83.40 | 62.92 | 78.25 | 63.07 | 82.42 | 61.98 | 81.68 | 61.80 | 76.30 | 65.91 | 79.04 | 64.13 |
| TC | 92.07 | 54.57 | 92.02 | 54.59 | 95.16 | 50.07 | 95.27 | 50.02 | 93.18 | 55.64 | 91.54 | 55.11 | 94.19 | 52.80 | 93.23 | 52.38 | 92.04 | 55.76 | 92.97 | 54.17 |
| E+BP+TC | 81.07 | 67.27 | 81.09 | 67.28 | 87.37 | 61.85 | 87.60 | 61.87 | 79.78 | 69.20 | 82.92 | 67.29 | 84.57 | 64.32 | 83.29 | 66.19 | 81.21 | 68.12 | 82.43 | 66.34 |
| TriScore | 65.18 | 75.36 | 64.97 | 75.36 | 78.99 | 66.67 | 79.20 | 66.85 | 72.27 | 72.07 | 70.80 | 72.39 | 73.95 | 70.57 | 72.66 | 71.54 | 65.75 | 75.81 | 68.85 | 73.76 |

Table 31: Adversarial OOD branch-combination results on ImageNet with ViT-B-16.

| Method | AA | | APGD-CE | | CW | | DF | | FGSM | | MPGD | | PGD | | SQUARE | | BIM | | PGD_BPDA | |
|---|---|---|---|---|---|---|---|---|---|---|---|---|---|---|---|---|---|---|---|---|
| | FPR@95 | AUROC | FPR@95 | AUROC | FPR@95 | AUROC | FPR@95 | AUROC | FPR@95 | AUROC | FPR@95 | AUROC | FPR@95 | AUROC | FPR@95 | AUROC | FPR@95 | AUROC | FPR@95 | AUROC |
| E | 92.13 | 59.70 | 92.06 | 59.69 | 94.34 | 51.74 | 94.44 | 51.62 | 89.88 | 64.17 | 88.64 | 69.83 | 93.11 | 55.97 | 91.01 | 64.62 | 91.76 | 60.07 | 92.41 | 58.27 |
| E+TC | 91.38 | 59.28 | 91.35 | 59.29 | 94.33 | 52.09 | 94.36 | 52.06 | 88.75 | 63.16 | 87.04 | 68.23 | 92.74 | 55.94 | 89.68 | 63.55 | 91.04 | 59.60 | 91.89 | 57.94 |
| E+BP | 84.11 | 61.39 | 84.30 | 61.42 | 92.58 | 53.33 | 92.69 | 53.43 | 81.65 | 64.32 | 79.65 | 68.16 | 88.86 | 58.02 | 85.68 | 63.80 | 84.44 | 61.92 | 86.66 | 59.98 |
| BP | 86.18 | 59.19 | 86.33 | 59.21 | 92.59 | 53.27 | 92.82 | 53.49 | 85.81 | 60.71 | 82.89 | 62.57 | 89.48 | 57.08 | 87.97 | 59.25 | 86.67 | 59.98 | 88.14 | 58.36 |
| BP+TC | 86.20 | 58.45 | 86.34 | 58.48 | 92.59 | 53.03 | 92.82 | 53.25 | 85.81 | 59.88 | 82.91 | 61.71 | 89.49 | 56.45 | 87.98 | 58.67 | 86.68 | 59.07 | 88.13 | 57.62 |
| TC | 88.18 | 55.31 | 88.20 | 55.28 | 94.23 | 51.52 | 94.17 | 51.72 | 86.51 | 58.01 | 80.24 | 61.96 | 92.43 | 53.60 | 85.78 | 57.33 | 88.78 | 55.67 | 90.99 | 54.49 |
| E+BP+TC | 88.32 | 60.37 | 87.98 | 60.41 | 94.00 | 52.41 | 94.04 | 52.65 | 83.39 | 64.67 | 79.92 | 70.18 | 91.68 | 56.77 | 84.93 | 65.16 | 88.10 | 60.74 | 90.25 | 58.79 |
| TriScore | 82.07 | 64.74 | 81.64 | 64.77 | 92.09 | 54.00 | 92.44 | 54.03 | 77.72 | 67.66 | 75.92 | 72.22 | 87.69 | 59.72 | 84.20 | 66.65 | 82.23 | 65.07 | 84.99 | 62.29 |

Table 32: Adversarial OOD branch-combination results on ImageNet with Swin-T.

| Method | AA | | APGD-CE | | CW | | DF | | FGSM | | MPGD | | PGD | | SQUARE | | BIM | | PGD_BPDA | |
|---|---|---|---|---|---|---|---|---|---|---|---|---|---|---|---|---|---|---|---|---|
| | FPR@95 | AUROC | FPR@95 | AUROC | FPR@95 | AUROC | FPR@95 | AUROC | FPR@95 | AUROC | FPR@95 | AUROC | FPR@95 | AUROC | FPR@95 | AUROC | FPR@95 | AUROC | FPR@95 | AUROC |
| E | 91.90 | 58.76 | 92.01 | 58.80 | 93.30 | 55.33 | 93.32 | 55.15 | 90.20 | 62.79 | 89.70 | 63.45 | 92.13 | 58.62 | 91.26 | 60.27 | 90.85 | 60.86 | 91.40 | 59.99 |
| E+TC | 91.76 | 58.66 | 91.90 | 58.70 | 93.27 | 54.76 | 93.29 | 54.69 | 90.31 | 61.97 | 89.43 | 62.09 | 92.02 | 57.96 | 90.81 | 59.58 | 90.66 | 60.46 | 91.47 | 59.27 |
| E+BP | 83.75 | 66.06 | 83.32 | 66.10 | 90.88 | 58.02 | 90.73 | 58.09 | 84.21 | 66.42 | 82.42 | 66.60 | 87.84 | 62.20 | 85.99 | 64.09 | 81.97 | 67.44 | 84.75 | 65.04 |
| BP | 78.83 | 67.45 | 78.99 | 67.44 | 89.52 | 57.66 | 89.74 | 57.99 | 82.28 | 64.99 | 79.16 | 64.50 | 85.92 | 61.55 | 85.69 | 62.90 | 79.71 | 67.91 | 82.41 | 64.90 |
| BP+TC | 83.12 | 61.60 | 82.80 | 61.58 | 91.33 | 53.97 | 91.45 | 54.13 | 85.56 | 60.49 | 85.07 | 58.49 | 88.92 | 56.86 | 88.33 | 57.25 | 83.12 | 62.06 | 86.06 | 59.38 |
| TC | 93.22 | 50.47 | 93.30 | 50.43 | 95.08 | 48.32 | 95.28 | 48.19 | 92.74 | 51.79 | 94.32 | 48.66 | 94.60 | 49.01 | 94.81 | 48.14 | 93.14 | 50.73 | 93.90 | 49.50 |
| E+BP+TC | 88.56 | 61.75 | 88.47 | 61.80 | 92.98 | 54.20 | 93.00 | 54.28 | 86.50 | 64.84 | 84.45 | 65.78 | 90.72 | 58.62 | 88.30 | 61.22 | 87.01 | 63.23 | 88.96 | 60.81 |
| TriScore | 74.60 | 69.83 | 74.60 | 69.85 | 89.60 | 58.01 | 89.60 | 58.28 | 76.72 | 68.70 | 73.77 | 68.46 | 84.50 | 62.87 | 82.98 | 65.19 | 74.59 | 70.70 | 79.11 | 67.20 |

