# OpenReview forum: "TriScore: Post-Hoc Out-of-Distribution Detection with Energy, Boundary Probes, and Transform Consistency"
_TMLR — Under review for TMLR_

### Review · Reviewer_o2tL · 2026-06-12

**Summary Of Contributions:**

This paper studies post-hoc out-of-distribution detection for frozen image classifiers, with a particular focus on classifier-level adversarial shifts. The authors propose TriScore, which combines three criteria: class-centered energy, Boundary-Probe fragility, and transform consistency. The energy branch measures abnormal logit evidence after subtracting an ID-estimated class-mean logit vector. The Boundary-Probe branch combines a top-k logit margin with the gradient of that margin at a late feature representation to approximate local decision fragility. The transform-consistency branch measures the Jensen–Shannon divergence between predictive distributions under deterministic image transformations. These three scores are standardized using ID validation data and combined through an input-dependent softmax gate based on their residuals above ID quantile thresholds. The intended contribution is to provide a practical post-hoc OOD detector that uses no OOD labels, auxiliary model, or classifier retraining, while improving robustness under non-adaptive adversarial shifts.

**Audience:**

Yes

**Audience Explanation:**

The topic is relevant to the TMLR audience, since it is related to out-of-distribution detection, adversarial robustness, post-hoc model evaluation, and deployment of frozen classifiers. The empirical observation that different OOD criteria fail differently across architectures and adversarial attacks is useful. The comprehensive comparison may also help researchers understand the practical tradeoffs among logit-based, feature-based, gradient-based, and transform-based detection signals.

The experiments and ablation studies are the strongest parts of the paper. The evaluation covers multiple architectures, attack families, clean OOD benchmarks, branch definitions, branch combinations, and computational costs. The paper also reports negative or mixed findings, such as the weak standalone performance of class-centered energy, the limited clean-OOD advantage of the gated fusion, and the architecture-dependent behavior of different branches. These results may interest readers even if the proposed method itself is methodologically incremental.

However, the current paper does not answer the more general and potentially more important question of how probe functions should be selected and combined. The paper begins with a manually chosen set of three criteria and then studies combinations within that restricted set. A more principled formulation could start from a broader bank of candidate probe functions, estimate their contribution and redundancy using OOD or adversarial validation data, select several complementary subsets, and then evaluate the selected detectors under the intended resource-restricted deployment setting. Such an approach would provide stronger insight into why particular criteria should be included.

**Claims And Evidence:**

No

**Claims Explanation:**

The paper provides extensive empirical evidence that the proposed combination performs well under the evaluated protocol. The experiments compare TriScore with 15 post-hoc OOD baselines across four ImageNet-scale backbones, standard near- and far-OOD datasets, and ten classifier-level adversarial attacks. The paper also includes branch-isolation experiments, branch-combination ablations, per-attack results, runtime comparisons, and additional evaluations on smaller datasets. These results support the narrower empirical claim that TriScore achieves strong average performance under the specific non-adaptive adversarial-shift protocol.

However, the evidence does not fully support the broader methodological motivation of the paper. A main concern is that the paper combines two related but different objectives: constructing a practical post-hoc detector under strict ID-only calibration, and identifying an effective detector for adversarially shifted data. The first objective motivates the restriction that no OOD or adversarial examples can be used during calibration. The second objective would naturally motivate selecting criteria based on their observed contribution under OOD and adversarial shifts. The method follows the first objective, while the main experimental claims emphasize the second. These directions are not sufficiently separated, which weakens the overall problem formulation.

Another concern is that the selection of the three criteria remains largely hand-designed. The paper gives a detailed intuitive explanation that energy captures global logit abnormality, Boundary-Probe captures local decision fragility, and transform consistency captures prediction instability under benign transformations. This explanation makes the selected criteria understandable, but it does not establish why these three should be preferred over other post-hoc OOD criteria. The paper does not provide a general selection principle, a systematic evaluation over a larger bank of probe functions, or an analysis showing that this particular combination is optimal or especially complementary.

The branch-combination ablations do not fully address this concern. The paper tests several subsets and fusion variants constructed from energy, Boundary-Probe, and transform consistency, including single branches, pairwise combinations, equal-weight fusion, and the proposed gated fusion. However, the candidate set has already been manually restricted to the three proposed criteria. The experiments therefore show which combination works best within the selected family, but they do not explain why this family should have been selected in the first place. Alternative combinations involving existing logit, feature, gradient, and geometry-based OOD scores are not systematically studied.

The paper also provides limited theoretical or mathematical analysis. The Boundary-Probe score is motivated through a first-order approximation of the feature-space boundary distance, but the approximation is not connected to a formal robustness guarantee. The paper does not establish that the three criteria are statistically or geometrically complementary, that their errors are sufficiently independent, or that the proposed residual gate should improve detection under unseen shifts. The softmax gate is intuitively reasonable, but its quantile thresholds and softness parameter remain heuristic design choices. Since the methodological novelty is already limited, stronger theoretical or analytical justification would be particularly important.

Finally, the proposed method appears to have limited methodological novelty. Energy-based detection, gradient or margin-based fragility measurements, test-time transform consistency, ID-based score standardization, and softmax-style score fusion are all closely related to established techniques. The individual modifications, including class-mean logit centering, the margin-gradient ratio, and residual-based gating, provide incremental refinements. The paper combines these components carefully and evaluates them extensively, but the overall contribution remains closer to a well-engineered integration of existing criteria than to a substantially new OOD detection principle.

**Requested Changes:**

The authors should provide a more principled justification for selecting the three proposed criteria. The current explanation is primarily intuitive and describes what each branch is intended to measure. The paper should either formulate a general criterion for selecting complementary probe functions or include a broader comparison of alternative probe combinations. This would help determine whether the reported improvement is specific to energy, Boundary-Probe, and transform consistency, or whether similar gains can be obtained by combining other existing OOD scores.

The authors should clarify the methodological novelty of each component. Class-centered energy, margin-gradient boundary estimation, Jensen–Shannon transform consistency, and residual-based softmax fusion should be positioned more carefully relative to existing energy, gradient, robustness, test-time augmentation, and score-ensemble methods. The paper should explain which parts constitute technically new contributions and which parts are adaptations or combinations of established ideas.

The authors should provide stronger theoretical or analytical support for the fusion mechanism. In particular, the paper should analyze why the selected criteria are expected to be complementary, under what conditions the residual gate improves over equal weighting, and how the quantile thresholds and softness parameter affect the detector. Formal guarantees may not be necessary, but a deeper analysis would make the method less dependent on heuristic motivation.

The authors should include stronger fusion baselines. In addition to comparing TriScore with individual post-hoc detectors and combinations of its own three branches, the paper should evaluate simple averaging, rank aggregation, calibrated linear fusion, or learned fusion using alternative existing OOD criteria. This comparison is necessary to distinguish the benefit of the specific TriScore design from the general benefit of combining multiple detector signals.

---

### Review · Reviewer_wHfr · 2026-06-17

**Summary Of Contributions:**

**Summary**:
This paper studies post-hoc OOD detection for a frozen classifier. The authors propose TriScore, which combines three signals extracted from the frozen model: a log-sum-exp energy score computed from the logits, a Boundary-Probe score that pairs a top-k logit margin with the gradient of that margin at a late feature layer, and a transform-consistency score computed as the Jensen–Shannon divergence between the model's output on the base view and on two mild deterministic views. The three scores are standardized using an ID dataset and then combined through an adaptive ID-residual gate. The authors evaluate TriScore in the OpenOOD framework on four backbones (ResNet-50, RegNet, ViT-B-16, and Swin-T), and report that it achieves the best mean AUROC among the baselines under the adversarial-shift benchmark.

**Strengths**:
Overall, the proposed method is simple, straightforward, and effective, and both the idea and the motivation came across clearly to me. The evaluation is also fairly broad: the four backbones cover both convolutional and transformer architectures, and the comparison spans 15 baselines. The ablation studies give useful insight into how much each component contributes to the final performance, though I think they could be taken a step further (see Weaknesses and Requested Changes). I also appreciated the Limitations section, where the authors are upfront about the single calibration run, the runtime cost, and so on; this makes the claims considerably easier to assess.

**Weaknesses**:
One of my main concern is on the paper's presentation. The paper is not the easiest to read: several notations are used before they are formally introduced, and a number of the tables are quite dense (I give specific suggestions under Requested Changes). Beyond presentation, I found that several design choices: k_m, the feature-layer location h, \epsilon, the crop scale and transform-pool composition, the quantile levels, and the softness s, are left unjustified and un-ablated, a few of them by the authors' own admission.

**Additional Comments:**

Please see my summary and my requested changes.

**Audience:**

Yes

**Audience Explanation:**

This paper presents a post-hoc method for OOD detection, which might be of interest to researchers working with OOD or generalization problems.

**Claims And Evidence:**

No

**Claims Explanation:**

**Claims that are not fully supported**:
My main concern is about these design choices. Because several of them are neither ablated nor justified (some, again, acknowledged as such by the authors), and because the results come from a single run with a few of the performance gaps being fairly small, the evidence for the claimed performance gains feels weaker than it could be. Concretely:
- k_m = 5 is used in place of a top-2 margin and described as "more stable", but without a supporting experiment;
- the late-feature hook location h is fixed with no study of where it is placed, how that placement affects performance, or why it should be the last layer before the head;
- the crop scale of 0.9 and the composition of the transform pool are not justified;
- the quantile levels (0.90/0.70/0.70) and the softness s = 0.6 are set as defaults, with the quantiles and s acknowledged as un-swept; and
- \epsilon is never assigned a value.

**Requested Changes:**

**Major**:

- Ablate the within-module design choices, or justify them with ID-only evidence and disclose how they were chosen: k_m (including the top-2 case), the feature-hook location h, the crop scale and transform-pool composition, the quantile levels, and s. Please also state the value of \epsilon and its sensitivity, and confirm explicitly that none of these constants were selected using OOD or adversarial data.
- Improve the presentation. It would help to define the model/dataset notation, D_ID, and the branch means/standard deviations/quantiles up front; to define T_geo before its first use; to move the complexity note out of the algorithm float; to state how \mu, \sigma, and the quantiles are computed; to specify how S(x) is turned into an OOD decision at a chosen operating point; and to fix the broken equation cross-references (for example, the "Eq. equation 7" rendering).
- Discuss, beyond the current single sentence, why TriScore is not the strongest method on clean OOD. For instance, which cue ends up diluted, and on which kinds of shift the specialized baselines do better.

**Minor**:

- In the appendix tables, it would help readers to make the best and second-best result in each column easier to spot (for example, by highlighting them).

---

### Review · Reviewer_nUVz · 2026-07-02

**Summary Of Contributions:**

This paper proposes TriScore, a post-hoc OOD detector for a fixed image classifier. The method combines three indicators: a class-centered energy score, a local boundary-sensitivity score, and a transform-consistency score. I read the method as a kind of multi-indicator residual estimator: one signal measures global logit scale, one measures local sensitivity to the classifier boundary, and one measures stability under simple transformations. The most distinctive part is the Boundary-Probe term. The authors define a top-k logit margin Δ(x), take its gradient with respect to a late feature representation h(x), and use a first-order proxy (with a gradient magnitude in the denominator). as a local estimate of distance to a decision boundary. The final Boundary-Probe score is large when this estimated distance is small. The three scores are then standardized on ID validation data and fused with an ID-only gating rule.

**Audience:**

Yes

**Audience Explanation:**

The paper studies a practically relevant setting: frozen classifiers, no retraining, and ID-only calibration. The idea of combining a global energy indicator, a local sensitivity indicator, and a stability indicator is simple and useful. I also think the Boundary-Probe idea will be interesting to readers, because it connects OOD detection to local conditioning of the classifier map.

That said, the paper would be stronger if it treated the Boundary-Probe term more like a numerical method: state the approximation clearly, discuss its invariances and failure modes, and test its sensitivity.

**Broader Impact Concerns:**

None.

**Claims And Evidence:**

Yes

**Claims Explanation:**

Yes, but only for the stated experimental setting. The empirical tables support the paper’s narrow claim: under the authors’ non-adaptive adversarial-shift benchmark, TriScore has the best mean AUROC across the four tested ImageNet-scale backbones. The ablations also support the claim that the proposed boundary and consistency indicators improve over simpler variants.

However, I think the gradient-based Boundary-Probe term needs a more careful treatment. The score is based on a first-order linearization of the classifier margin in feature space. This is a reasonable idea, but it is also coordinate-dependent. If the feature representation is rescaled or reparameterized, the gradient norm changes, even when the classifier function is essentially the same. So the quantity delta^2/|| grad_h delta||^2 is not an intrinsic distance to the boundary, but a distance in feature coordinates. That is not necessarily wrong, but the paper should be more explicit that this is a heuristic sensitivity indicator, not a geometric boundary distance.

here is also a conditioning issue. The method divides by norm of gradient squared + a guard, and the behavior can change depending on the feature scale, the chosen hook location, and the size of the stabilizing guard. From a numerical-analysis perspective, I would want diagnostics: distribution of gradient norms, sensitivity to the hook location, and whether the denominator is ever close to the floor.

The adversarial evaluation also interacts with this gradient term. Since the attacks are generated against the classifier and not against TriScore, they do not test whether the Boundary-Probe score itself can be fooled. A detector-aware attack would need to differentiate through a quantity involving grad_h delta, which means the attack may involve second-order information or a BPDA/stop-gradient approximation. This feels important for robustness against adaptive attacks.

So my view is: the evidence supports the empirical claims as stated, but the broader robustness interpretation is not yet fully established.

**Requested Changes:**

1. Clarify the interpretation of the boundary probe, or promote it to something more meaningful than it is.
2. Discuss feature coordinate dependence of the boundary probe score. Maybe include an experiment showing sensitivity to the hook location or feature normalization.
3. Report gradient diagnostics.
4. Analyze whether the boundary probe is just detecting hard examples. I'm curious on whether the probe is adding an OOD-specific piece of information or just measuring uncertainty/conditioning in the classifier.
5. Since TriScore uses gradients as part of the score, the key missing experiment is an adaptive attack against the detector itself. For example, generate perturbations that fool the classifier while also minimizing TriScore. If exact differentiation through BP is too expensive because of second-order terms, then a BPDA or stop-gradient approximation should be reported.
6. Maybe make robustness language more precise? The experiments don't support robustness against a detector-aware attack.
7. Add uncertainty estimates.
8. TriScore isn't obviously the cleanest OOD method. Maybe the framing should point this out.